# A thousand inversions to determine Quantifying European SF<sub>6</sub> emissions from 2005 to 2021 using a large inversion ensemble

Martin Vojta<sup>1,2</sup>, Andreas Plach<sup>1,3</sup>, Rona L. Thompson<sup>4</sup>, Pallav Purohit<sup>5</sup>, Kieran Stanley<sup>6</sup>, Simon O'Doherty<sup>6</sup>, Dickon Young<sup>6</sup>, Joe Pitt<sup>6</sup>, Jgor Arduini<sup>7,8</sup>, Xin Lan<sup>9,10</sup>, and Andreas Stohl<sup>1</sup>

Correspondence: Martin Vojta (martin.vojta@univie.ac.at)

#### Abstract.

We determine European emissions of sulfur Sulfur hexafluoride (SF<sub>6</sub>) is a highly potent and long-lived greenhouse gas whose atmospheric concentrations are increasing due to human emissions. In this study, we determine European SF<sub>6</sub> emissions from 2005 to 2021 using a large ensemble of atmospheric inversions. To assess uncertainty, we systematically vary key inversion parameters across 986 sensitivity tests and apply a Monte Carlo approach to randomly combine these parameters in 1,003 additional inversions. Our analysis focuses on high-emitting countries with robust observational coverage — UK, Germany, France, and Italy — while also examining aggregated EU-27 emissions.

SF<sub>6</sub> emissions declined across all studied regions except Italy, largely attributed to EU F-gas regulations (2006, 2014), however, national reports underestimated emissions: (i) UK emissions dropped from 65 (±1368 (47-77) t yr<sup>-1</sup> in 2008 to 20 (±619 (15-26) t yr<sup>-1</sup> in 2018, aligning with the reports from 2018 onward; (ii) French emissions fell from 88 (±3778 (51-117) t yr<sup>-1</sup> (2005) to 51 (±2835 (19-54) t yr<sup>-1</sup> (2021), exceeding reports by 7388%; (iii) Italian emissions fluctuated (31-67-25-48 t yr<sup>-1</sup>), surpassing reports by 88107%; (iv) German emissions declined from 166 (±41182 (155-251) t yr<sup>-1</sup> (2005) to 95 (±1197 (88-104) t yr<sup>-1</sup> (2021), aligning reasonably well with reports; (v) EU-27 emissions decreased from 484 (±213403 (335-501) t yr<sup>-1</sup> (2005) to 255 (±58225 (191-260) t yr<sup>-1</sup> (2021), exceeding reports by 4020%. A substantial drop from 2017 to 2018 mirrored the trend in southern Germany, suggesting regional actions were taken as the 2014 EU regulation took effect. Our sensitivity tests highlight the crucial role of dense monitoring networks in improving inversion reliability. The UK system expansions (2012, 2014) significantly enhanced result robustness, demonstrating the importance of comprehensive observational networks in refining emission estimates.

<sup>&</sup>lt;sup>1</sup>Department of Meteorology and Geophysics, University of Vienna, Vienna, Austria

<sup>&</sup>lt;sup>2</sup>Department of Chemistry, University of Crete, Crete, Greece

<sup>&</sup>lt;sup>3</sup>Department of Meteorology, Stockholm University and Bolin Centre for Climate Research, Stockholm, Sweden

<sup>&</sup>lt;sup>4</sup>NILU, Kieller, Norway

<sup>&</sup>lt;sup>5</sup>International Institute for Applied Systems Analysis (IIASA), Laxenburg, Austria

<sup>&</sup>lt;sup>6</sup>School of Chemistry, University of Bristol, Bristol, United Kingdom

<sup>&</sup>lt;sup>7</sup>Institute of Atmospheric Sciences and Climate, National Research Council, Bologna, Italy

<sup>&</sup>lt;sup>8</sup>Department of Pure and Applied Sciences, University of Urbino "Carlo Bo", Urbino, Italy

<sup>&</sup>lt;sup>9</sup>CIRES, University of Colorado, Boulder, CO, USA

<sup>&</sup>lt;sup>10</sup>NOAA Global Monitoring Laboratory, Boulder, CO, USA

Sulfur hexafluoride (SF<sub>6</sub>) is a fluorinated gas (F-gas), which has the highest global warming potential (GWP) of all known greenhouse gases (GHG), with current estimates of 18,400, 24,700, and 29,800 for 20-, 100-, and 500-year time horizons, respectively (WMO, 2022). Even more concerning, estimates of its atmospheric lifetime range between 850 and 1,280 years (WMO, 2022), implying that SF<sub>6</sub> from past, present, and future anthropogenic emissions will accumulate in the atmosphere and will warm the climate for thousands of years.

Since the late 1990s, global  $SF_6$  mole fractions have almost tripled, from 4.2 ppt in 1998 to 11.4 ppt in 2023 (Lan et al., 2024b), while global atmospheric growth rates have more than doubled, from 0.20 t  $yr^{-1}$  in 1998 to 0.41 t  $yr^{-1}$  in 2023 (Lan et al., 2024b). Radiative forcing increased from 2.4 in 1998 to 6.2 mW m<sup>-2</sup> in 2022. If the current global emission trend continues,  $SF_6$  radiative forcing could increase up to 70 mW m<sup>-2</sup> by the end of the century (Hu et al., 2023).

Due to its high stability, SF<sub>6</sub> is used mainly as an insulator for electric equipment in the power industry (e.g. IEEE, 2012; Koch et al., 2018; Cui et al., 2024), with emissions occurring during equipment leakage, failures, maintenance, and decommissioning. It is also used in the metal industry as a blanketing gas (e.g. Maiss and Brenninkmeijer, 1998), as a cover gas in magnesium production and processing (Bartos et al., 2007; Ottinger et al., 2015), for semiconductor manufacturing for equipment cleaning and plasma etching (e.g. Cheng et al., 2013), and in the past it was even used to fill sports shoes (Pedersen, 2000) and car tires (Schwaab, 2000). In the 1990s, especially in Western Europe, SF<sub>6</sub> was used to fill double-glazed windows (e.g. Schwarz, 2005), which still represents a substantial European emission source (United Nations Framework Convention on Climate Change, 2023).

Since the late 1990s, global  $SF_6$  mole fractions have almost tripled, from 4.2 ppt in 1998 to 11.4 ppt in 2023 (Lan et al., 2024b), while global atmospheric growth rates have more than doubled, from 0.20 t yr $^{-1}$  in 1998 to 0.41 t yr $^{-1}$  in 2023 (Lan et al., 2024b). Radiative foreing increased from 2.4 in 1998 to 6.2 mW m $^{-2}$  in 2022. If the current global emission trend continues,  $SF_6$  radiative foreing could increase up to 70 mW m $^{-2}$  by the end of the century (Hu et al., 2023).

SF<sub>6</sub> was regulated under the Kyoto Protocol, where it is listed as one of the six categories of major GHGs (United Nations Framework Convention on Climate Change, 1997). To meet the Kyoto Protocol's targets, the EU passed regulation No. 842/2006 (EU, 2006), setting rules for the containment, recovery, use, and reporting of fluorinated gases. It banned the use of SF<sub>6</sub> in vehicle tires (starting in 2007) and in large-scale magnesium die-casting (starting in 2008), as well as in soundproof windows and footwear. The EU's 2014 regulation (No.517/2014, EU, 2014) further restricted SF<sub>6</sub> use, requiring leak detection systems for electrical switchgear by 2017 and banning it from recycling magnesium alloys by 2018. The new 2024 F-gas regulation mandates the phase-out of F-gases in medium-voltage switchgear by 2030, high-voltage switchgear by 2032, and prohibits SF<sub>6</sub> use for switchgear maintenance by 2035, unless reclaimed or recycled (EU, 2024).

A key aspect of the Kyoto Protocol was the implementation of a robust system for monitoring GHG emissions, requiring Annex-I countries (industrialized countries) to submit annual reports to the United Nations Framework Convention on Climate Change (UNFCCC), including SF<sub>6</sub>. These reports are almost exclusively calculated by so-called bottom-up methods, where statistical data on economic production and consumption are combined with source-specific emission factors to estimate

national emissions. The Emissions Database for Global Atmospheric Research (EDGAR) and the Greenhouse Gas and Air Pollution Interactions and Synergies (GAINS) model also provide bottom-up inventories of SF<sub>6</sub> emissions. However, due to inherent uncertainties associated with bottom-up methods, there is a strong demand for independent verification (e.g. Weiss et al., 2021), which can be achieved through top-down approaches, such as inverse modeling (e.g. Leip et al., 2017). In an inversion approach, atmospheric observations are used together with an atmospheric transport model to optimize the emissions.

Several inversion studies have investigated  $SF_6$  emissions, however, limited research has specifically focused on the European continent. The global inversion study by Rigby et al. (2010) estimated total European  $SF_6$  emissions from 2004 to 2008, distinguishing between emissions from reporting and non-reporting countries. Ganesan et al. (2014) estimated  $SF_6$  emissions for 2012 in well-monitored countries, including Germany, France, and the UK. Their estimates indicated higher emissions than those officially reported to the UNFCCC. Brunner et al. (2017) used four independent inverse models to estimate European  $SF_6$  emissions in 2011. Their results were 47% higher than the UNFCCC reports, with Germany identified as the largest emitter. Simmonds et al. (2020) used three different inversion systems to estimate total  $SF_6$  emissions from western Europe between 2013 and 2018, with one of the systems extending its analysis to cover 2007-2018. Their calculated emissions ranged from comparable to significantly higher than the reported values. Their work also suggested substantial  $SF_6$  emissions in southwest Germany. In the UK's annual report to the UNFCCC, Manning et al. (2022) presented inversion results for  $SF_6$  emissions in both the UK and northwest Europe, revealing a downward trend in both regions. The global inversion study by Vojta et al. (2024) provided an annual  $SF_6$  time series for the aggregated EU-27 emissions, between 2005 and 2021. They found a decline in  $SF_6$  emissions, with a significant drop in 2018, which they attributed to the impact of the EU's 2014 F-gas regulation.

While recent studies have employed regional high-resolution inversions to constrain SF<sub>6</sub> emissions in China (An et al., 2024) and the U.S. (Hu et al., 2023), there is no recent high-resolution regional study examining the trend of SF<sub>6</sub> emissions in Europe. This research endeavors to bridge a significant gap in our understanding of European SF<sub>6</sub> emissions. We adopt the methodology established by Vojta et al. (2024), adapting it for a high resolution (0.25x0.25°) inversion covering all of Europe. Utilizing the same datasets, we quantify SF<sub>6</sub> emissions across the continent for the period 2005 to 2021. While Vojta et al. (2024) primarily investigated the influence of different a priori inventories, this study delves deeper by conducting a comprehensive sensitivity and uncertainty analysis. We systematically examine the impact of a wide range of parameters on our inversion results, enabling a more robust quantification of overall uncertainties and a thorough investigation of the sensitivity to individual inversion components.

#### 80 2 Methods

### 2.1 Measurement data

In this study, we utilize the same global observational dataset as employed by Vojta et al. (2024), where a detailed description is available. Therefore, we only provide a brief overview here. The dataset is based on globally distributed atmospheric observations of SF<sub>6</sub> dry-air mole fractions collected between 2005 and 2021. It includes continuous on-line measurements, instantaneous flask sample data from surface stations, and observations from mobile platforms. The measurements were

contributed by various independent organizations such as the National Oceanic and Atmospheric Administration (NOAA) and the Advanced Global Atmospheric Gases Experiment (AGAGE) international network. Continuous surface measurements were averaged over 3-hour intervals and all observations were standardized to the SIO-2005 calibration scale (for more details see Vojta et al., 2024). It is noteworthy that the number of available European on-line monitoring stations increased over the study period. While at the beginning only 5 such stations were available (Bialystok: BIK, Jungfraujoch: JFJ, Mace Head: MHD, Zeppelin: ZEP, Zugspitze-Schneefernerhaus: ZSF), the monitoring network in Western Europe significantly expanded with the addition of UK observations from Ridge Hill (RGL) and Tacolneston Tall Tower (TAC) in 2012 and further from Bilsdale (BSD) and Heathfield (HFD) in 2014. Figure S1 provides an overview of all the ground-based measurements globally, while Fig. 1 shows the stations in Europe.

To determine the influence of data selection criteria on our results, we created eight different subsets. 1) We used the entire global dataset (presented in Vojta et al., 2024), and 2) we selected a subset comprising only stations located in and around Europe (created a European subset by excluding on-line stations outside Europe (BRW, CGO, COI, GSN, HAT, IZO, MLO, NWR, RPB, SMO, SPO, SUM, THD; see Fig.S1) while retaining the European sites<sup>1</sup> (BIK, BRM, BSD, CMN, HFD, JFJ, MHD, RGL, TAC, ZEP, and ZSF; see Fig. 1). For each of these two Note that the stations SUM in Greenland and IZO in Tenerife are geographically closest to the European inversion domain. For these global and European datasets, we further refined the selection by: a) retaining only night observations (00:00 - 06:00) at mountain stations and afternoon observations (12:00 - 18:00) at all other sites for continuous monitoring stations; b) creating a data subset that excludes mountain stations, and c) generating a subset that omits low-frequency measurements and data from moving platforms, retaining only high-frequency surface observations. Table S1 provides the number of observations used from each dataset for each year, whereas Tab. S2 shows the availability of online measurements within and outside Europe.

# 2.2 Emission sensitivities

95

100

105

115

We use the Lagrangian particle dispersion model (LPDM) FLEXPART 10.4 (Pisso et al., 2019) in backward mode to simulate the atmospheric transport of SF<sub>6</sub>, tracing its movement from the measurement locations back to the emission sources. We neglect loss processes, given that SF<sub>6</sub> is almost inert up to the middle stratosphere. For every observation, we release 50,000 virtual particles continuously over a 3-hour interval from the measurement site, tracking their trajectories backward in time for 50 days. The average time spent by these particles in a given emission grid cell determines the sensitivity of the observation to emissions from that specific grid cell. These simulated emission sensitivities form the basis for the atmospheric inversion. We run FLEXPART on a European domain (15 °W-40 °E, 30 °N-72 °N) with an output resolution of 0.25 °× 0.25 ° and on a global domain with an output resolution of 1.0 °× 1.0 °, both with 18 vertical layers of 0.1, 0.5, 1, 2, 3, 4, 5, 7, 9, 11, 13, 15, 17, 20, 25, 30, 40, and 50 km above ground level (agl) interface heights. The emission sensitivities were calculated solely for the lowest layer, ranging from 0 to 100 meters agl, where almost all emissions occur. We utilize hourly ECMWF ERA5<sup>2</sup> wind fields

<sup>&</sup>lt;sup>1</sup>We initially still kept the globally distributed flask measurements and observations from moving platforms to improve the baseline optimization

<sup>&</sup>lt;sup>2</sup>ERA5 is the fifth generation of the European Centre for Medium-Range Weather Forecasts (ECMWF) atmospheric reanalysis, providing comprehensive global climate and weather data from January 1940 to the present.

**Figure 1.** Locations of the observation stations in and around Europe. Stations with continuous surface measurements (BIK, BSD, BRM, BSD, CMN, HFD, JFJ, MHD, RGL, TAC, ZEP, ZSF) are represented with red triangles, while surface flask measurements (BAL, BSC, CIB, HPB, HUN, LMP, MHD, OBN, OXK, MHD, PAL, STM, TAC, WIS, ZEP) are shown with black dots.

(Hersbach et al., 2018) to drive the FLEXPART simulations. Specifically, we use  $0.25 \,^{\circ} \times 0.25 \,^{\circ}$  resolution wind fields for the European domain and  $0.5 \,^{\circ} \times 0.5 \,^{\circ}$  wind fields for the global domain, both with 137 vertical levels. Figure 2 shows the simulated annual averaged emission sensitivities for the years (a) 2005 and (b) 2018. In 2018, Europe - particularly northwestern Europe - shows significantly higher emission sensitivities compared to 2005. This increase is largely due to the expansion of the observation network in the UK. As a result, northwestern Europe, including major emitters such as Germany, France, and the UK, became well-monitored, suggesting that substantial improvements in emission estimates through the inversion can be expected over time.

#### 2.2.1 Baseline

Using the emission sensitivities simulated by FLEXPART, we can link the mole fractions at the receptor to emissions occurring within 50 days of the backward tracking. However, emissions preceding the 50-day period can not directly be captured with these backward simulations. Nevertheless, they still have to be considered when comparing the modeled mole fraction values with the observations. Therefore, all these emission contributions are aggregated in a so-called baseline which must be added to the modeled emission contributions. We apply a Global-Distribution-Based (GDB) method (Vojta et al., 2022) to calculate the baseline, directly from a 3D global mole fraction field. For this, the endpoints of the FLEXPART back-trajectories, are used to determine an observation's sensitivity to the mole fractions at the end of the 50-day simulation period. These sensitivities are simply obtained by dividing the number of trajectories ending in a specific grid cell by the total number of trajectories calculated, as loss processes are omitted. We then multiply the sensitivities with globally 3D gridded SF<sub>6</sub> mole fractions at the time of particle termination and integrate the product over the entire globe. As for the 3D SF<sub>6</sub> field, we employed the data set

**Figure 2.** Simulated emission sensitivities for the years 2005 and 2018 in Europe. The displayed values represent the annual sum of FLEXPART calculations. As a result, sites with frequent online observations carry more weight than those relying on flask measurements or observations from mobile platforms.

created by Vojta et al. (2024). Finally, the contributions from emissions occurring during the 50-day FLEXPART simulation period but outside of the European domain are added to the baseline. For a more detailed description of the GDB method and the simulation of the mole fraction fields please see Vojta et al. (2022) and Vojta et al. (2024).

# 2.3 A priori emissions





We create seven different European  $SF_6$  a priori emission fields with  $0.25 \times 0.25^{\circ}$  resolution for our inversion domain and for the period 2005 to 2021 that are based on three different bottom-up sources: GAINS, the annual national emission reports to the UNFCCC, and the bottom-up estimates from EDGAR.

- GAINS: We created two a priori emission fields based on the GAINS inventory (Purohit and Höglund-Isaksson, 2017), which is detailed in Vojta et al. (2024). The inventory is available at 0.5° resolution globally and at 0.1° resolution for a European subset covering the EU-27, Iceland, Norway, Switzerland, and the UK. For the first field, we started with a global inventory at a resolution of (GS), we re-gridded the global 0.5° and regridded these data to a finer resolution of inventory to 0.25° for over the European domain (GS), using by interpolation. For the second field (GS-HR), we used a high-resolution emission dataset available at 0.1×0.1° for the EU-27, Iceland, Norway, Switzerland, and the UK. We also regridded this dataset the higher-resolution European dataset, aggregated it to 0.25°, and combined it with the first dataset(GS-HR)global dataset. While both fields thus share the same resolution (0.25° over Europe), the information content differs: GS is interpolated from coarser data, whereas GS-HR retains detail from the original high-resolution European inventory.
- Reports to the UNFCCC: We utilize the total national SF<sub>6</sub> emissions reported annually to the UNFCCC (United Nations Framework Convention on Climate Change, 2023) and distribute these emissions within each country's borders on a 0.25°×0.25° grid, based on two different proxy datasets: (1) gridded population density (CIESIN, 2018) (UP), or (2) nightlight remote sensing data (Elvidge et al., 2021) (UN). <sup>3</sup>
- EDGAR: We use the newly updated, 0.1°×0.1°-gridded annual SF<sub>6</sub> emission inventory EDGARv8.0 (EDGAR, 2023; Crippa et al., 2023) and regrid it to 0.25°×0.25° resolution (E8). In addition, we also utilize the national annual total emissions provided by the previous version EDGARv7.0 (EDGAR, 2022; Crippa et al., 2021), and distribute those totals according to (1) gridded population density (CIESIN, 2018) (E7P) or (2) night light remote sensing (Elvidge et al., 2021) (E7N).

To account for contributions from emissions occurring during the 50-day FLEXPART simulation period but outside the European domain, we utilize the global a priori emission fields generated by Vojta et al. (2024). These fields were calculated using the same methodology as our European a priori fields but at a coarser resolution of  $1.0 \times 1.0^{\circ}$ . Note that this approach results in a single global coarse GAINS inventory.

<sup>&</sup>lt;sup>3</sup>Emissions of non-Annex I countries, that fall within our inversion domain but are not further investigated, were estimated proportionally to their national electricity generation as described in Vojta et al. (2024).

**Figure 3.** Seven a priori emission fields, shown as an average over the study period 2005-2021: (a) GS (GAINS), (b) GS\_HR (GAINS high resolution), (c) UP (UNFCCC reports - population density distribution), (d) UN (UNFCCC reports - night light remote sensing distribution), (e) E8 (EDGARv8), (f) E7P (EDGARv7 - population density distribution), and (g) E7N (EDGARv7 - night light remote sensing distribution). The table in the bottom right corner provides an overview of the ensemble.

Figure 3 shows the seven generated European a priori emission fields averaged across the study period from 2005 to 2021. Overall, these emission fields display a relatively similar spatial pattern, especially when compared to the significantly larger differences observed in the global patterns of the bottom-up SF<sub>6</sub> inventories (see Vojta et al., 2024). All European inventories show high SF<sub>6</sub> emissions in central Europe, with Germany being the largest emitter. Notably, a priori emissions are particularly high in western Germany, especially in the area around Cologne. The EDGAR and UNFCCC inventories also highlight substantial emissions in Berlin, which are less prominent in the GAINS inventories. In France, the UNFCCC and EDGAR inventories concentrate emissions in the Paris region, whereas GAINS indicates more dispersed emissions. For the UK, all inventories show the highest emissions in London, with elevated values also occurring in other large cities such as Liverpool, Manchester, and Birmingham. In Italy, the EDGAR inventories estimate higher a priori emissions compared to those from GAINS and UNFCCC. Substantial emissions occur also in the Moscow region, particularly for the EDGAR- and GAINS-based inventories. A detailed discussion on the differences among the a priori inventories and their influence on inversion results are provided in Appendix B.

#### 2.4 Inversion method





We use the Bayesian inversion framework FLEXINVERT+ to find an optimized estimate for European  $SF_6$  emissions based on a priori emissions (Sect. 2.3), atmospheric observations (Sect. 2.1), and atmospheric transport (Sect. 2.2). The framework minimizes the cost function J:

$$\mathbf{J}(\mathbf{x}) = \frac{1}{2} (\mathbf{x} - \mathbf{x}_p)^T \mathbf{B}^{-1} (\mathbf{x} - \mathbf{x}_p) + \frac{1}{2} (\mathbf{H}\mathbf{x} - \mathbf{y})^T \mathbf{R}^{-1} (\mathbf{H}\mathbf{x} - \mathbf{y}),$$
(1)

where  $\mathbf{x}$  and  $\mathbf{x}_p$  represent the state vector and its a priori values, respectively;  $\mathbf{y}$  represents the mole fraction enhancements with respect to the baseline,  $\mathbf{H}$  represents the atmospheric transport operator,  $\mathbf{B}$  is the a priori error covariance matrix, and  $\mathbf{R}$  is the observation error covariance matrix. From a Bayesian point of view,  $\mathbf{J}$  represents the negative logarithm of the a posteriori probability distribution, derived using Bayes' theorem (e.g., Tarantola, 2005). The minimum of the cost function, therefore, defines the maximum of the a posteriori probability distribution, and provides the most probable emission estimate (a posteriori emissions). The analytic solution to minimize  $\mathbf{J}$ , reads:

$$\hat{\mathbf{x}} = \mathbf{x}_p + \mathbf{G}(\mathbf{y} - \mathbf{H}\mathbf{x}_p) \tag{2}$$

with the defined gain matrix **G**:



$$\mathbf{I90} \quad \mathbf{G} = \mathbf{B}\mathbf{H}^T (\mathbf{H}\mathbf{B}\mathbf{H}^T + \mathbf{R})^{-1} \tag{3}$$

The a posteriori emission error covariance matrix,  $\hat{\mathbf{B}}$ , can also be derived analytically using:

$$\hat{\mathbf{B}} = \mathbf{B} - \mathbf{G}\mathbf{H}\mathbf{B} \tag{4}$$

For SF<sub>6</sub>, positive fluxes are expected over land, but the inversion may still yield negative a posteriori values in some grid cells. To correct this, we apply the truncated Gaussian method of (Thacker, 2007), which enforces non-negativity as an inequality constraint. The adjusted fluxes  $\hat{\mathbf{x}}'$  are calculated as

$$\hat{\mathbf{x}}' = \hat{\mathbf{x}} + \hat{\mathbf{B}}\mathbf{P}^T \left(\mathbf{P}\hat{\mathbf{B}}\mathbf{P}^T\right)^{-1} (\mathbf{c} - \mathbf{P}\hat{\mathbf{x}}),$$

where  $\hat{\mathbf{x}}$  is the original a posteriori estimate,  $\hat{\mathbf{B}}$  the a posteriori error covariance matrix,  $\hat{\mathbf{P}}$  the operator identifying violations, and  $\hat{\mathbf{c}}$  the constraint vector.

A detailed description of the inversion framework FLEXINVERT+ is provided by Thompson and Stohl (2014), and its application to SF<sub>6</sub> in Vojta et al. (2024). Although we also use observations from outside the European domain and perform FLEXPART simulations globally (with a European nest), the inversions are regional; that is, emissions are optimized only within Europe. Following the inversion process, national emission totals are calculated by aggregating the a posteriori emissions within the respective grid cells of the corresponding country, employing a national identifier grid (CIESIN, 2018).

## 2.5 Inversion sensitivity studies

A key challenge in inverse modeling is accurately determining the uncertainties of the optimized emissions. Traditionally, the uncertainties in inversion-derived emissions are based on Gaussian error statistics within a Bayesian framework, often relying on a single inversion setup. However, many aspects of the inversion process are based on assumptions and expert

judgments. One example is the error covariance matrix used for the a priori emissions. Both the magnitude of this uncertainty as well as its spatiotemporal correlation are usually not available from the bottom-up inventories, and the assumption of a Gaussian distribution is not always well justified. Several studies (e.g., Bergamaschi et al., 2015; Brunner et al., 2017; Chevallier et al., 2019; Locatelli et al., 2013) have demonstrated that the range of emissions derived from different inversion configurations can be significantly larger than the uncertainties calculated by individual inversions. Therefore, in this study, we examine the sensitivity of the inversion results to various inversion settings. Initially, we define 58 distinct inversion settings by systematically varying key parameters, starting from a reference inversion. These settings For the reference inversion, parameter choices were informed by a set of preliminary runs, in which we evaluated chi-squared statistics, and by values reported in previous studies. The settings of the sensitivity tests are applied to each of the 17 years in the study period (2005–2021), resulting in a total of 986 inversions (58 × 17). Our sensitivity tests include:

- *a priori* emissions: We use 7 different a priori emissions emission fields based on the inventories of GAINS, the UNFCCC reports and EDGAR (see Sect.2.3).
- *a priori* emission uncertainties: In each grid cell, the a priori emission uncertainty is calculated as a fraction of its respective emission value. We test 4 different settings with fractions of 30%, 50%, 70%, and 100%. Furthermore, different minimum absolute values for the emission uncertainty are tested, controlling the freedom of the algorithm to adjust emissions in grid cells with small a priori values. The seven minimum values tested are  $5 \times 10^{-14}$ ,  $1 \times 10^{-13}$ ,  $5 \times 10^{-13}$ ,  $1 \times 10^{-12}$ ,  $5 \times 10^{-12}$ ,  $1 \times 10^{-11}$ , and  $5 \times 10^{-11}$  kg m<sup>-2</sup> h<sup>-1</sup>.
  - spatial a priori emission uncertainty correlations: FLEXINVERT+ uses an exponential decay function to account for spatial emission uncertainty correlations. We test different spatial scale lengths of 50, 100, 250, 500, 1000 km, as well as a setup with no spatial correlation.
    - observation datasets: We test all of the eight observation subsets described in Sect.2.1.




- observation uncertainties: FLEXINVERT+ assumes a diagonal observation error covariance , matrix and we test different configurations of this uncertainty. The observation uncertainty includes the transport model error projected into the observation space, which is assumed to be the dominant part. Initially, we test constant values of 0.02, 0.04, 0.06, 0.08, and 0.1 ppt. However, it is likely that the model error varies both spatially and temporally. To account for the spatial dependencies, uncertainty estimates are often based on model residuals (the difference between observed and simulated mole fractions) at the measurement stations (e.g. Stohl et al., 2009; Henne et al., 2016). Therefore, we also test two different approaches: (i) using the RMSE between prior modeled and observed values, averaged per station, to determine the observation error, and (ii) estimating the model error from the standard deviation of the a posteriori error distribution

<sup>&</sup>lt;sup>4</sup>Omitting the off-diagonal elements of the observation error covariance matrix could potentially lead to an underestimation of the total observation uncertainty, resulting in an over-weighting of observations. This could especially be relevant for high-frequency observations, driving results further away from the a priori emissions. However, we reduce this risk by averaging the observations and by verifying through chi-squared statistics that the assumed uncertainties remain consistent with the data.

through a series of initial inversion runs. In idealized experiments, it has been demonstrated that incorporating temporally varying, flow-dependent uncertainty can enhance the accuracy of emission estimates (Steiner et al., 2024). This transport model ensemble approach, however, requires a lot of resources and lies beyond the scope of our study.

- baseline optimization: FLEXINVERT+ includes an option for baseline optimization, where spatially aggregated contributions are optimized on a coarse grid. Firstly, we test different resolutions for the coarse grid, where the global field is divided into 8, 4, and 2 latitude bands, with northern edges at [-60°, -30°, -15°, 0°, 15°, 30°, 60°, 90°], [-30°, 0°, 30°, 90°], [0°, 90°], respectively. We also run inversions where we optimize the global field at onceone scaling factor for the whole global field. Additionally, we tried different temporal baseline optimization intervals of 15, 30, 45 and 60 days.
   Furthermore, we test various baseline uncertainty values set to 0.0001, 0.0003, 0.0005, 0.0007, 0.0009, 0.001, 0.01, 0.1, and 1 ppt, and run an inversion without any baseline optimization.
  - emission grid: We use emission grids with varying cell sizes, determined by aggregating cells with low emission contributions based on emission sensitivities and a priori emissions (for further details, see Thompson and Stohl, 2014). We test grid configurations with 588, 744, 1,992, 2,781, 4,248, 5,370, and 7,229 cells, each configuration kept constant over time. Additionally, we implement three dynamic setups where the grid configuration changes each year, with the number of cells ranging from (i) 2,781 to 5,916, (ii) 3,645 to 6,599, and (iii) 4,151 to 7,229.

The set-up of the reference inversion and an overview of all tested inversion configurations are listed in Tab. 1.

## 2.6 Inversion uncertainties





While the sensitivity studies give provide insight into how different parameter settings influence the inversion results, the overall uncertainties of the inversion are determined by all these parameters simultaneously. Therefore, in order to quantify the uncertainties of the inversion results accurately quantify the inversion uncertainties, one must apply random variations of these parameters. However, testing all possible combinations is infeasible due to the vast number of permutations. To address this, we employ a Monte Carlo method (e.g. Metropolis et al., 1953) to randomly select and combine inversion parameters, generating an additional 59-member ensemble (1,003 inversions). Since the uncertainty distribution of the input parameters is not known, parameters are sampled either continuously from a uniform distribution within a defined range normal distribution or discretely from a set of predetermined values. The Monte Carlo sampling of the parameter space is performed independently for each parameter. The final selection of parameter ranges for ensemble construction is based on the results of our sensitivity tests. A detailed overview of the ensemble configuration is provided in Table ??. Please note that for this approach, we excluded parameters to which the inversion showed negligible sensitivity, as they were unlikely to significantly impact the uncertainty. Based on the sensitivity tests, we further refined our parameter settings by narrowing the ranges for the a priori emission uncertainty, the spatial a priori uncertainty correlation length, and observation uncertainty (see TableS3, and the choices are discussed in Appendix J. To assess the representativeness of our ensemble and the robustness of our findings, we constructed three additional independent Monte Carlo ensembles using the same parameter ranges: (i) another 59-member ensemble with different parameters compared to the initial ensemble (E59, see Tab. ??). All inversions employed an emission grid of 558 cells,

Table 1. Reference inversion set-up and overview of all inversion configurations used in the sensitivity tests

| Inversion aspect                                                        | Reference set-up    | Tested configuration                                       | Number of tests <sup>a</sup> |
|-------------------------------------------------------------------------|---------------------|------------------------------------------------------------|------------------------------|
| A priori emissions inventory                                            | GS                  | GS, GS-HR, UP, UN, E8, E7P, E7N                            | 6 (+1)                       |
| A priori emissions uncertainty [%]                                      | 50                  | 30, 50, 70, 100                                            | 3 (+1)                       |
| Minimal a priori emission value [kg $\mathrm{m}^{-2}~\mathrm{h}^{-1}$ ] | $1 \times 10^{-13}$ | $5 \times 10^{-14}, 1 \times 10^{-13}, 5 \times 10^{-13},$ | 6 (+1)                       |
|                                                                         |                     | $1 \times 10^{-12}, 5 \times 10^{-12}, 1 \times 10^{-11},$ |                              |
|                                                                         |                     | $5 \times 10^{-11}$                                        |                              |
| A priori uncertainty decorrelation distance [km]                        | 250                 | no correlation, 50, 100, 250, 500, 1000                    | 5 (+1)                       |
| Observation dataset                                                     | Global              | Global                                                     | 7 (+1)                       |
|                                                                         |                     | Global: excluding mountain stations,                       |                              |
|                                                                         |                     | Global: night/afternoon selection,                         |                              |
|                                                                         |                     | Global: high-frequency surface stations,                   |                              |
|                                                                         |                     | Europe                                                     |                              |
|                                                                         |                     | Europe: excluding mountain stations,                       |                              |
|                                                                         |                     | Europe: night/afternoon selection,                         |                              |
|                                                                         |                     | Europe: high-frequency surface stations                    |                              |
| Observation uncertainty [ppt]                                           | 0.6-0.06            | 0.02, 0.04, 0.06, 0.08, 0.1,                               | 6 (+1)                       |
|                                                                         |                     | standard deviation (a posteriori distribution),            |                              |
|                                                                         |                     | RMSE (a priori distribution)                               |                              |
| Baseline optimization: grid resolutions [#]                             | 4                   | 1, 2, 4, 8                                                 | 3 (+1)                       |
| Baseline optimization: temporal window [days]                           | 30                  | 15, 30, 45, 60                                             | 3 (+1)                       |
| Baseline optimization: uncertainty [ppt]                                | 0.1                 | no optimization, 0.0001, 0.0003, 0.0005,                   | 9 (+1)                       |
|                                                                         |                     | 0.0007, 0.0009, 0.001, 0.01, 0.1, 1                        |                              |
| number of gridcells [#]                                                 | 3,645-6,599         | 588; 744; 1,992; 2,781; 4,248; 5,370; 7,229;               | 9 (+1)                       |
|                                                                         |                     | 2,781-5,916; 3,645-6,599; 4,151-7,229                      |                              |

<sup>&</sup>lt;sup>a</sup> (+1) represents the reference inversion setup, which adds one test to the total number for each parameter. However, as the reference inversion is only conducted once, it is only counted once in the overall number of tests.

and the baseline was optimized in 8 latitudinal bands, using a time window of S4), (ii) an ensemble with half the original size (30 days and a baseline uncertainty of 0.1 pptmembers, E30, see Tab. S5), and (iii) an ensemble with double the original size (118 members, E118, see Tab. S6). The results of these ensembles were then evaluated against those of the original ensemble.

#### 3 Results and discussion

## 3.1 Inversion increments, error reduction, and a posteriori emission distribution

Figure 4 presents (a) the inversion increments (a posteriori minus a priori), (b) the relative error reduction, calculated for each grid cell based on the a priori and a posteriori emission uncertainties as  $1 - \frac{a \text{ posteriori uncertainty}}{a \text{ priori uncertainty}}$ , (c) the a posteriori emission distribution, and (d) the a posteriori emissions uncertainty (as calculated using Eq. 4). The results are shown for the reference inversion settings, averaged over the years 2005–2021. Strongly negative increments (Fig. 4a) are found in northern Germany, particularly around Cologne, where a priori emissions are very high (see also Fig. 3). The inversion further reveals large positive increments in southwestern Germany. Other positive increments are evident in France, Italy, the UK, and Russia, whereas negative increments are observed in Switzerland, parts of Scotland, and Israel.

Consistent with the distribution of emission sensitivities (Fig. 2), the largest error reductions (Fig. 4b) are concentrated in central Europe, particularly in well-monitored countries such as the UK, Germany, and Switzerland. Additional areas with notable error reduction include northwestern France and northernmost Italy, Moscow and Israel. Notice that the elevated error reductions in Moscow and Israel are likely a consequence of the relatively high a priori uncertainties in these regions, which give the algorithm more flexibility to adjust emissions. However, since these areas are poorly covered by the observation network, the a posteriori emissions may still be considered unreliable despite the notable error reduction.

Figure 4c reveals particularly high emissions in southwestern Germany, aligning with the findings of Simmonds et al. (2018), who also reported an emission maximum in this region. We also obtain elevated emissions in southern UK, northern and southeastern France, and northern Italy. A posteriori emissions are also high in Moscow and Israel, which, however, show large a posteriori uncertainties, despite the notable uncertainty reductions there (Fig. 4d). As discussed in the Appendix B, the Russian a posteriori emissions are highly sensitive to the choice of a priori emissions, leading to unstable results. In the following sections, we therefore focus on the high-emitting European countries with better observational coverage: the UK, Germany, France, and Italy.

Table S7 and Fig. S3 demonstrate the statistical improvements at all continuous surface stations, with the sites TAC, HFD, RGL, and BSD showing the largest improvements, thereby highlighting the importance of the UK network expansion.

#### 3.2 Results of the sensitivity tests






The results of all performed sensitivity tests are detailed in the Appendix, organized according to various aspects of the inversion process to assess the sensitivity of the results to each specific setting. Generally, the sensitivities to the different tested settings vary between different years and regions, however, there is one common feature: The better a region is monitored by the observation network, the smaller is the sensitivity to the inversion setting. This is especially apparent in the well-monitored countries like Germany and the UK, where the inversion results are extremely stable across all tested cases (Fig. 11).

In our spectrum of sensitivity tests, inversion results were most sensitive to changes in the spatial correlation length of the a priori emission uncertainty (Fig. D1) and to changes in the baseline uncertainty (Fig. G4). The results were also sensitive to the magnitude of the a priori emission uncertainties (Fig. C1/C3) and observation errors (Fig. F1), with greater sensitivity in poorly monitored areas and minimal sensitivity in well-monitored regions. Additionally, the inversion results were moderately sensitive to the choice of the observation dataset (Fig. E1). Furthermore, our tests suggest that optimizing the baseline across two or more latitudinal bands can lead to substantial differences compared to optimizing the global field with a single scalar (Fig. G1). In contrast, changing the temporal interval for baseline optimization, ranging between 15 and 60 days, had almost no impact on the results (Fig. G3). Also, changing the number of optimized emission grid cells had minimal impact on the results when considering national or European emission totals (Fig. H2). This finding is particularly noteworthy, as computational time is heavily influenced by the number of inversion grid cells.

**Figure 4.** (a) Emission increments from the inversion (a posteriori - a priori), (b) relative error reduction, (c) a posteriori emission distribution, and (d) a posteriori uncertainty obtained for our reference inversion and averaged over all years of the study period 2005-2021.

## 3.3 A posteriori emission emission ensemble

Building on the sensitivity tests, we employed a Monte Carlo ensemble presented in Tab. ?? §3 and calculated the a posteriori emissions for all different settings. The ensemble mean (Fig. 5a) closely resembles the a posteriori emissions of the reference inversion (Fig. 5b), with the largest differences observed in Moscow and Israel, and generally more pronounced discrepancies

**Figure 5.** A posteriori emissions: (a) ensemble mean, calculated as the average of all a posteriori emissions from the Monte Carlo ensemble over the study period (2005–2021), and (b) difference between the ensemble mean a posteriori emissions and the a posteriori emissions obtained from the reference inversion.

in larger cities. Note that the ensemble spread (Fig. ?? S2) results in significantly larger uncertainties than the analytically derived uncertainties from the reference inversion (Fig. 4d), which likely fail to capture the full extent of the actual uncertainty.

#### 3.4 Regional emission time series



For the regional emission time series, the inversion results of all members of the Monte Carlo ensemble are shown in Fig. ?? and S4 and final results are presented in Tab. ??. Our final results S8, which are defined as ensemble averages across the full set of inversions, with a 2-\sigma 2.5th 97.5th percentile uncertainty range for each year. Doubling or halving the ensemble size yielded consistent posterior results, suggesting that the ensemble is sufficiently large to represent the prescribed uncertainty distributions (see Fig. S5).

Figure 6 shows the a posteriori SF<sub>6</sub> emission time series for (a) the United Kingdom, (b) Germany, (c) France, (d) Italy, and (e) the EU-27. In the UK, SF<sub>6</sub> emissions declined from  $41 \pm 1338 \pm 31-46$  t yr<sup>-1</sup> in 2005 to  $20 \pm 619 \pm 15-26$  t yr<sup>-1</sup> by 2021 (Fig. 6a: black solid line), with a substantial drop from  $65 \pm 1368 \pm 1000$  (47-77) t yr<sup>-1</sup> in 2008 to  $20 \pm 1000$  (15-26) t yr<sup>-1</sup> in 2018, corresponding to an average annual decrease of  $3 \pm 3.2$  t yr<sup>-1</sup>. The substantial decrease in emissions observed after 2008 is likely a result of the 2006 EU F-gas regulations, with most bans coming into effect in 2008. Although inversion results exceed the reported values (dashed red line) by an average of 5069% between 2005 and 2017, they align closely from 2018 onward. While our results are slightly higher than those the estimates of Ganesan et al. (2014) in 2012, they agree within the uncertainties with the estimates of Brunner et al. (2017) in 2011. Our results and Brunner et al. (2017) in

Figure 6. Annual emission time series for (a) the United Kingdom, (b) Germany, (c) France, (d) Italy, and (e) the EU-27. The solid black lines represent the average a posteriori emissions across all performed inversions, and a 2-σ uncertainty shaded areas indicate the 2.5th-97.5th percentile range for each year is indicated by gray shading. The red dashed line indicates the UNFCCC reported emissions, while results from previous studies are shown with colored markers. The vertical grey gray lines indicate the times when additional observational data from the expansion of the UK network became available.

2011, they are in excellent agreement with the results of Manning et al. (2022) for the whole study period, particularly from 2012 onward, when uncertainties also become significantly smaller. We find equally good agreements when comparing with the a posteriori emissions for northwestern Europe from Manning et al. (2022) (Fig. ??\$6). These good agreements are a particularly noteworthy result, as the inversion system used by Manning et al. (2022) differs significantly from ours. Their approach employs the InTEM inversion technique (Manning et al., 2011, 2021), with a priori emissions uniformly distributed across the country, large a priori emission uncertainties, and inversion intervals of 1 and 3 months (Fig. 6, InTEM 1mth, InTEM 3mth).








In Germany, our results show a decline in emissions from  $\frac{166(\pm 41182)}{166(\pm 41182)}$  tyr<sup>-1</sup> in 2005 to  $\frac{91(\pm 2886)}{166(\pm 41182)}$  (66-109) t  $yr^{-1}$  in 2013. Afterwards, emissions increased significantly, peaking at  $\frac{205 (\pm 42) \text{ t yr}^{-1}}{199}$  (172-241) in 2017. This was followed by a sharp drop to  $\frac{105 \pm 15109}{105 \pm 15109}$  (97-125) t yr<sup>-1</sup> in 2018, after which emissions stabilized. Over the entire study period, emissions decreased from  $\frac{166(\pm 41182 (155-251) \text{ t yr}^{-1} \text{ in } 2005 \text{ to } \frac{95(\pm 1197 (88-104) \text{ t yr}^{-1} \text{ in } 2021. \text{ Overall, the}}{2005 \text{ to } \frac{95(\pm 1197 (88-104) \text{ t yr}^{-1} \text{ in } 2021. \text{ overall, the}}{2005 \text{ to } \frac{95(\pm 1197 (88-104) \text{ t yr}^{-1} \text{ in } 2021. \text{ overall, the}}{2005 \text{ to } \frac{95(\pm 1197 (88-104) \text{ t yr}^{-1} \text{ in } 2021. \text{ overall, the}}{2005 \text{ to } \frac{95(\pm 1197 (88-104) \text{ t yr}^{-1} \text{ in } 2021. \text{ overall, the}}{2005 \text{ to } \frac{95(\pm 1197 (88-104) \text{ t yr}^{-1} \text{ in } 2021. \text{ overall, the}}{2005 \text{ to } \frac{95(\pm 1197 (88-104) \text{ t yr}^{-1} \text{ in } 2021. \text{ overall, the}}{2005 \text{ to } \frac{95(\pm 1197 (88-104) \text{ t yr}^{-1} \text{ in } 2021. \text{ overall, the}}{2005 \text{ to } \frac{95(\pm 1197 (88-104) \text{ t yr}^{-1} \text{ in } 2021. \text{ overall, the}}{2005 \text{ to } \frac{95(\pm 1197 (88-104) \text{ t yr}^{-1} \text{ in } 2021. \text{ overall, the}}{2005 \text{ to } \frac{95(\pm 1197 (88-104) \text{ t yr}^{-1} \text{ in } 2021. \text{ overall, the}}{2005 \text{ to } \frac{95(\pm 1197 (88-104) \text{ t yr}^{-1} \text{ in } 2021. \text{ overall, the}}{2005 \text{ to } \frac{95(\pm 1197 (88-104) \text{ t yr}^{-1} \text{ in } 2021. \text{ overall, the}}{2005 \text{ to } \frac{95(\pm 1197 (88-104) \text{ t yr}^{-1} \text{ t yr}^$ German a posteriori emissions align well with the values reported to the UNFCCC, however, the inversion results reveal distinct emission trends during specific time periods that are not reflected in the reported data. Our results for Germany agree well with three of the four inversions performed in Brunner et al. (2017), but give much lower emissions than those estimated by Ganesan et al. (2014). However, their high estimates were likely a result of the use of excessive German a priori emissions ( $\sim$ 650 t yr<sup>-1</sup>), which were based on the EDGAR v4.2 inventory. Although their German a posteriori emissions were substantially lower than their a priori values, the inversion likely could not fully correct the huge bias present in this version of the EDGAR inventory. As mentioned in Sect. 3.1, the inversion reveals notable regional differences between southern and northern Germany, with significant negative increments in the North and substantial positive increments in the South, especially in the Southwest (Fig. 7a). To further investigate these regional variations, we examine the annual emission trends separately for the North (Fig. 7b) and the South (Fig. 7c), with the division between the two regions at 51°N. In the north, SF<sub>6</sub> emissions decreased substantially, from  $\frac{76}{12}(\pm \frac{27}{79}-152)$  t yr<sup>-1</sup> in 2005 to  $\frac{25}{27}(\pm \frac{5}{22}-33)$  t yr<sup>-1</sup> by 2021. Note that this decrease in emissions is comparable to the reduction observed in the UK. In contrast, the southern emission trend follows a similar pattern to that of Germany as a whole, including a peak of  $\frac{180}{166} (\pm 49125 - 205)$  t yr<sup>-1</sup> in 2017, followed by a sharp decline to 93 ( $\pm 1497 - 110$ )  $t \text{ vr}^{-1} \text{ in } 2018.$ 

In France, a posteriori emissions declined from  $88 \pm 3778 \pm 117$  t yr<sup>-1</sup> in 2005 to  $51 \pm 2835 \pm 19-54$  t yr<sup>-1</sup> in 2021, with an average annual decrease of -1-1.2 t yr<sup>-1</sup>. This decline, however, remains within the uncertainty range, which is particularly large at the beginning of the study period and decreases after 2014. Our results exceed the reported values throughout the whole study period by 7388% on average, while they are in good agreement with the Ganesan et al. (2014), and the lower estimates of Brunner et al. (2017) and Ganesan et al. (2014).

For Italy, our inversion results exhibit large uncertainties in certain years, likely due to limited observational constraint in the central and southern regions. Over the study period, annual a posteriori emissions do not show a clear trend, varying between 31 and 67-25 and 48 t yr<sup>-1</sup>; however, they exceed the values reported to the UNFCCC by 88107% on average. Our results are within the range of estimates calculated in Brunner et al. (2017), which show a comparable level of uncertainty.

**Figure 7.** Inversion increments from the reference inversion averaged over the period 2005-2021, with the dashed line indicating our separation of northern and southern Germany (a). Annual emission time series for (b) northern Germany (>51°N) and (c) southern Germany (<51°N). The solid black lines represent the average a posteriori emissions across all performed inversions, and  $\frac{a}{a} - \frac{2}{\sigma} - \frac{c}{d} = \frac{2}{\sigma} + \frac{$ 

For the aggregated emissions of the EU-27 countries, our results show a decrease in a posteriori emissions from 484(±213403 (335-501) t yr<sup>-1</sup> in 2005 to 255(±58225 (191-260) t yr<sup>-1</sup> in 2021, with a substantial emission drop from 469(±144396 (311-490) t yr<sup>-1</sup> in 2018 to 291 (±622017 to 256 (216-303) t yr<sup>-1</sup> in 2017. 2018. While until 2017 our results are on average 4028% higher than the reported values, they align well with the reports from 2018 onward. Our results are very similar to the recent estimates of the global SF<sub>6</sub> inversion study of Vojta et al. (2024) -after 2012. This is not surprising, as we use the same dataset, atmospheric transport model, and inversion framework. In specific years Before 2012, our values slightly deviate are slightly higher from those in Vojta et al. (2024), which we attribute to the improved resolution of our study, the baseline optimization in 8 latitudinal bands, and the definition of our a posteriori emissions as averages over a large inversion ensemble. Our uncertainty intervals, defined as the 2-σ-2.5th–97.5th percentile uncertainty range across the performed inversions, are much wider than those reported in Vojta et al. (2024). Their uncertainty intervals, in contrast, were based on the minimum and maximum uncertainty limits across inversion results using only six different a priori emission inventories.


Note that the temporal pattern of EU-27 a posteriori emissions closely resembles the German pattern after 2012, as Germany is the largest European SF<sub>6</sub> emitter. The high emissions in Southern Germany (Fig. 7c), in particular, seem to have a large influence on the total EU-27 emission variations. We interpret the decline in SF<sub>6</sub> emissions as a consequence of the EU F-gas regulations introduced in 2006 (EU, 2006) and 2014 (EU, 2014). As suggested by Vojta et al. (2024), the sharp drop in EU-27 emissions from 2017 to 2018 might indicate an immediate effect of the 2014 regulation, which mandated that new electrical

switchgear be put into service starting in 2017 and banned the use of  $SF_6$  in recycling magnesium die-casting alloys from 2018. It seems that strong actions were taken particularly in south Germany when the 2014 regulation came into effect.

The vertical solid dashed gray lines in Fig. 6 indicate the times when additional observational data from the expansion of the UK network became available (RGL: March 2012, TAC July 2012, HFD: January 2014, BSD: February 2014). Consistent with our sensitivity studies, we observe that the additional observations noticeably reduce emission uncertainties in the UK. Similarly, this effect is observed in Germany, particularly in the north (see Fig. 7). However, the large southern emissions in 2016 and 2017 led to elevated uncertainties, primarily due to the use of different observational datasets (see Sect. Appendix E), making them an exception to this trend. Our tests cover a broad range of key inversion settings; however, additional factors such as alternative atmospheric transport models, wind field data, or inversion frameworks could lead to further deviations from our results. Nevertheless, the excellent agreement of the emissions in the UK and northwestern Europe (Fig. 6a and Fig. ??\$6) with those reported by Manning et al. (2022), particularly after the network expansion, suggests might suggest that, with a dense monitoring network, inversion results remain stable even when these factors change.

## 4 Conclusions






In this study, we estimated European  $SF_6$  emissions from 2005 to 2021, focusing on the largest emitters - the United Kingdom, Germany, France, Italy - and the aggregated EU-27 emissions. We conducted an extensive ensemble of 987 inversions to test the sensitivity of the results to various settings within the inversion framework. Building on this, we performed an additional 1003 inversions, using Monte Carlo methods to randomly select and combine inversion parameters, allowing us to quantify the uncertainties in the inversion results. The key findings of our study are as follows:

- We observe a decline in SF<sub>6</sub> emissions across most of the studied countries, as well as in the aggregated EU-27 emissions, over the period from 2005 to 2021. We interpret these declining emissions as a direct consequence of the EU F-gas regulations in 2006 and 2014. While our results are consistent with previous inversion studies, they indicate clearly that European countries generally underreport their SF<sub>6</sub> emissions to the UNFCCC.
- In the UK, SF<sub>6</sub> emissions decreased from 41 (±1338 (31-46) t yr<sup>-1</sup> in 2005 to 20 (±619 (15-26) t yr<sup>-1</sup> by 2021, with a considerable decrease from 65 (±1368 (47-77) t yr<sup>-1</sup> in 2008 to 20 (±619 (15-26) t yr<sup>-1</sup> in 2018, corresponding to an average annual decrease of -3-3.2 t yr<sup>-1</sup>. While the inversion results are, on average, 5069% higher than the values reported to UNFCCC prior to 2018, they align closely for the most recent investigated years, 2018–2021.
- Germany is the largest SF<sub>6</sub> emitter in Europe. Over the study period, emissions decreased from 166 (±41182 (155-251) t yr<sup>-1</sup> in 2005 to 95 (±1197 (88-104) t yr<sup>-1</sup> in 2021, aligning relatively well with UNFCCC-reported values. Our results suggest that emission inventories overestimate emissions in northern Germany and underestimate emissions in southern Germany (division at 51°N). Emissions in northern Germany declined from 76112 (±2779-152) t yr<sup>-1</sup> in 2005 to 2527 (±522-33) t yr<sup>-1</sup> in 2021, while emissions in southern Germany showed a distinct peak of 180166 (±49125-205) t yr<sup>-1</sup> in 2017, followed by a sharp decline to 93 (±1497-110) t yr<sup>-1</sup> in 2018.

- In France, a posteriori emissions decreased from  $\frac{88 (\pm 3778 (51-117) \text{ t yr}^{-1} \text{ in } 2005 \text{ to } \frac{51 (\pm 2835 (19-54) \text{ t yr}^{-1} \text{ in } 2021, on average exceeding the reported values by } \frac{73}{2}88\%$ .
- In Italy, annual a posteriori emissions show no clear trend, varying between 31 and 67 25 and 48 t yr<sup>-1</sup> throughout the study period, On average, emissions exceeded the reported UNFCCC values by 88107%.
- For the aggregated emissions of the EU-27 countries, our results show a decrease in a posteriori emissions from 484 (±213403 (335-501) t yr<sup>-1</sup> in 2005 to 255 (±58225 (191-260) t yr<sup>-1</sup> in 2021, with a substantial emission drop from 469 (±144396 (311-490) t yr<sup>-1</sup> in 2017 to 291 (±62256 (216-303) t yr<sup>-1</sup> in 2018. On average, our results are 4028% higher than the reported values before 2018, however, after the drop in 2018, they align better with the reported values from 2018 to 2021. As noted by Vojta et al. (2024), this drop is likely a direct result of the 2014 regulation, which mandated that new electrical switchgear containing SF<sub>6</sub> be put into service starting in 2017 and prohibited the use of SF<sub>6</sub> in recycling magnesium die-casting alloys from 2018. Additionally, we notice that the drop closely mirrors the decline in emissions in southern Germany over the same period, suggesting that strong actions were likely taken there when these regulations came into force.
- Our large ensemble of sensitivity tests shows that as the observational coverage in a region increases, the inversion results become less sensitive to the various a priori settings that are subject to uncertainty. This becomes especially apparent in countries like Germany and the UK, where the inversion results stabilize substantially following the expansion of the British observation network. The good agreement of emissions in the UK and northwest Europe after 2014 with Manning et al. (2022) further suggests that factors not tested in this study such as alternative atmospheric transport models, meteorological data driving the models, or different inversion frameworks become less significant when a dense monitoring system is in place. It also demonstrates the considerable potential of inverse modeling to provide reliable emission estimates and underscores the importance of extending the existing network (e.g. Weiss et al., 2021; Leip et al., 2017).

In addition, our sensitivity tests, described in detail in the Appendix, reveal the following:

- Inversion results demonstrate high sensitivity to the choice of spatial correlation length for the a priori emission uncertainty, ranging from 0 to 1000 km. While the optimal correlation length depends on the specific problem, a range of 50 and around 250 km appears to produce relatively stable results a good compromise between providing sufficient constraint on emissions and maintaining the inversion's ability to resolve regional emission patterns. In contrast, correlation lengths of 500 km, 1000 km, or the absence of correlation led to substantial differences. When emission uncertainties are assumed to be entirely uncorrelated, the inverse problem becomes relatively ill-determined.
   However, excessively large correlation lengths prevent the inversion's ability to capture regional emission patterns.
  - Inversion results were also significantly influenced by the choice of baseline uncertainty, particularly within the range of 0 to 0.0003 ppt. Although increasing the uncertainty further led to additional changes, the effect gradually diminished,

stabilizing between 0.01 and 0.1 ppt. We recommend ensuring the baseline uncertainty is not underestimated, as this can significantly impact the results.

- Optimizing the baseline using two or more latitudinal bands most likely yields better results than optimizing the entire field with a single scalar. Therefore, we recommend optimizing the baseline in at least two latitudinal bands, particularly for species such as SF<sub>6</sub>, which exhibit a strong latitudinal gradient and large interhemispheric differences. In contrast, the choice of temporal interval for baseline optimization (ranging from 15 to 60 days) had minimal impact on the results.
  - The number of optimized emission grid cells, ranging from 588 to 7,229, had minimal impact on the obtained national total emissions. Given that the computational time for an inversion is strongly influenced by the number of grid cells optimized, we recommend conducting prior sensitivity tests related to the grid configuration. This approach could help conserve significant computational resources using a coarser grid where appropriate.

- Results were sensitive to variations in both a priori emission uncertainties and observation errors, with greater sensitivity
  in poorly monitored areas and minimal impact in well-monitored regions. We recommend conducting sensitivity tests
  on these uncertainties to improve the accuracy of uncertainty estimates in inversion results.
- Inversion results showed moderate sensitivity to the choice of the observation dataset and a priori emission fields. Again, we suggest using multiple a priori emission datasets and observational datasets to improve the reliability of uncertainty estimates in the inversion results.

Our study indicates that regulations, such as those implemented by the EU for F-gases, can have a significant positive impact on regional GHG emissions. It will be interesting to observe how the EU's new 2024 F-gas regulation will further reduce European SF<sub>6</sub> emissions in the future. Considering the substantial regional emission reductions observed, Europe could serve as a role-model for effectively reducing SF<sub>6</sub> emissions. Similar regulations would be crucial in other regions for mitigating global SF<sub>6</sub> emissions (Vojta et al., 2024; An et al., 2024). Furthermore, expanding observation networks - similar to the dense British network - should be a top priority, as this would greatly reduce uncertainties in top-down emission estimates derived from inverse modeling. These improved estimates could then be incorporated into national reports, as already done by Switzerland, the UK, and Australia (e.g. Rypdal et al., 2005; Leip et al., 2017), substantially enhancing our understanding of GHG emissions.

. The FLEXINVERT+ code (described by Thompson and Stohl, 2014), along with configuration files, is provided at https://doi.org/10.25365/phaidra.648. The FLEXPART 10.4 source code (described by Pisso et al., 2019) is accessible at https://doi.org/10.5281/zenodo.3542278. FLEXPART 8-CTM-1.1 and its user guide can be freely downloaded from https://doi.org/10.5281/zenodo.1249190 (Henne et al., 2018). Daily global SF<sub>6</sub> mole fraction fields from the re-analysis (2005–2021) are available at https://doi.org/10.25365/phaidra.489.

All links and references to the atmospheric mole fractions used in this study are detailed in Vojta et al. (2024) and are repeated here for convenience: AGAGE data: https://data.ess-dive.lbl.gov/view/doi%3A10.15485%2F1909711 (Prinn et al., 2023); Heathfield Tall

Tower data: https://catalogue.ceda.ac.uk/uuid/df502fe4715c4177ab5e4e367a99316b (Arnold et al., 2019); Bilsdale Tall Tower data: https:// 480 //catalogue.ceda.ac.uk/uuid/d2090552c8fe4c16a2fd7d616adc2d9f (O'Doherty et al., 2019); Zeppelin mountain data: https://ebas-data.nilu. no/Pages/DataSetList.aspx?key=4548F59E3CBD48E0A505E8968BD268EB (2005-2010 EBAS, 2024); NOAA/GML Chromatograph for Atmospheric Trace Species (CATS) program: https://doi.org/10.7289/V5X0659V (all stations, hourly data, Dutton and Hall, 2023); Monte Cimone, Cape Ochiishi, Izaña, Ridge Hill, Zugspitze-Schneefernerhaus: https://doi.org/10.50849/WDCGG SF6 ALL 2022 (di Sarra et al., 2022); Atmospheric SF<sub>6</sub> Dry Air Mole Fractions from the NOAA GML Carbon Cycle Cooperative Global Air Sampling Network: 485 https://doi.org/10.15138/p646-pa37 (Lan et al., 2024a); NOAA Global Greenhouse Gas Reference Network provided flask-air PFP sample measurements of SF<sub>6</sub> at Tall Towers and other Continental Sites https://doi.org/10.15138/5R14-K382 (Andrews et al., 2022); Atmospheric Sulfur Hexafluoride Dry Air Mole Fractions from the NOAA GML Carbon Cycle Aircraft Vertical Profile Network https://doi.org/10. 15138/39HR-9N34: (McKain et al., 2022); NOAA ObsPACK SF<sub>6</sub> data: https://doi.org/10.15138/g3ks7p (NOAA Carbon Cycle Group ObsPack Team, 2018); IAGOS-CARIBIC Aircraft measurements: https://zenodo.org/records/10495039 (Schuck and Obersteiner, 2024); 490 NOAA/ESRL/GMD/HATS Trace Gas Measurements from Airborne Platforms: https://gml.noaa.gov/aftp/data/hats/airborne/ (Elkins et al., 2020); NOAA Atmospheric Carbon and Transport - America aircraft measurements: https://doi.org/10.3334/ORNLDAAC/1575 (Sweeney et al., 2018). For the observations at BIK (Popa et al., 2010), BRM (Rust et al., 2022), GSN (Kim et al., 2012), and HAT (Saikawa et al., 2012) we refer to E. Popa <epopa2@yahoo.com>, S. Reimann <stefan.reimann@empa.ch>, S. Park <sparky@knu.ac.kr>, and T. Saito <saito.takuya@nies.go.jp>, respectivley.

- . MV, AP, and AS designed the study. MV performed the FLEXPART, and FLEXINVERT+ simulations and made the figures. RT helped with the FLEXINVERT+ setup and simulation issues. KS, SO, DY, JP, JA, and XL provided atmospheric observation data. PP provided GAINS emission estimates. MV wrote the text with input from AP, RT, PP, KS, SO, DY, JA, JP, XL, and AS
  - . The authors declare that they have no conflict of interest.
- . We sincerely thank the entire AGAGE team for their contribution of measurement data, including Christina M. Harth, Jens Mühle,
  Peter K. Salameh, and Ray F. Weiss (Scripps Institution of Oceanography, UCSD); Paul B. Krummel, Paul J. Fraser, and Paul Steele
  (CSIRO Environment); Hsiang-Jui (Ray) Wang (Georgia Institute of Technology); Martin K. Vollmer and Stefan Reimann (Empa,
  Swiss Federal Laboratories for Materials Science and Technology); and Chris Rene Lunder and Ove Hermanson (NILU). We are
  especially grateful to all station managers and operators. The operation of AGAGE stations at Mace Head, Trinidad Head, Cape
  Matatula, Ragged Point, and Kennaook / Cape Grim has been supported by NASA (USA) grants to MIT (grant numbers NAG5505 12669, NNX07AE89G, NNX11AF17G, NNX16AC98G, and 80NSSC21K1210) and SIO (grant numbers NNX07AE87G, NNX07AF09G,
  NNX11AF15G, NNX11AF16G, NNX16AC96G, NNX16AC97G, and 80NSSC21K1201). The UK measurements were funded by the
  UK Government Department for Energy Security and Net Zero (DESNZ) under contracts TRN1028/06/2015, TRN1537/06/2018 and
  TRN5488/11/2021 to the University of Bristol and through the National Measurement System at the National Physical Laboratory; NOAA

(USA) under contract number 1305M319CNRMJ0028 to the University of Bristol for Ragged Point; CSIRO, BoM, DCCEEW, and RRA (Australia); NILU (Norway); KNU (Korea); CMA (China); NIES (Japan); and Urbino University (Italy).








For Jungfraujoch, funding from the Swiss Federal Office for the Environment (FOEN) is acknowledged for the HALCLIM/CLIMGAS-CH project, as well as support from the Swiss National Science Foundation for ICOS (Integrated Carbon Observation System). Observations are further supported by the International Foundation High Altitude Research Stations Jungfraujoch and Gornergrat (HFSJG). The halocarbon measurements at Zeppelin Observatory are supported by the Norwegian Environment Agency. We thank CSIRO and the Bureau of Meteorology in Australia for their longstanding support of Kennaook / Cape Grim and its science program. Operations of the Gosan station on Jeju Island, South Korea, were supported by a grant (no. RS-2023-00229318) from the National Research Foundation of Korea funded by the South Korean government (MSIT).

We also extend our thanks to the NOAA Global Monitoring Laboratory for making their data accessible, with special recognition to Geoff Dutton, James W. Elkins, Fred Moore, Eric Hintsa, Dale Hurst, Bradley Hall, Kathryn McKain, Stephen Montzka, John Bharat Miller, Colm Sweeney, Ed Dlugokencky, Xin Lan, Arlyn Andrews, David Nance, and Christina M. Harth. We are also grateful to key collaborators Le Huang (EC) and Kenneth Davis (PSU). Data from the Southern Great Plains (SGP) were collected by Sébastien Biraud (Lawrence Berkeley National Laboratory, USA) with support from the U.S. Department of Energy's Office of Biological and Environmental Research under contract no. DE-AC02-05CH11231, as part of the Atmospheric Radiation Measurement (ARM) and Environmental System Science (ESS) programs.

We also acknowledge the contributions of many individuals and institutions who shared their observational data, including Takuya Saito (National Institute for Environmental Studies, Japan), Sunyoung Park, Lee Gawon, (Kyungpook National University - operations of the Gosan station on Jeju Island, South Korea were supported by the National Research Foundation of Korea grant funded by the Korean government MSIT no. 2020R1A2C3003774), Luciana Vanni Gatti (Instituto Nacional de Pesquisas Espaciais) and Emanuel Gloor (University of Leeds) - Pantanal SF<sub>6</sub> measurements were supported by NERC grant NE/N015657/1, Emilio Cuevas (State Meteorological Agency, Spain), Dan Say (University of Bristol), Jgor Arduini (University of Urbino), Cedric Couret (German Environment Agency), Elena Popa (Utrecht University), Satoshi Sugawara (Miyagi University of Education, Japan), Armin Rauthe (German Weather Service) and Jonathan Williams (Max Planck Institute for Chemistry), Tanja Schuck (Goethe University Frankfurt), and Florian Obersteiner (Karlsruhe Institute of Technology).

The computational results were partly achieved using the Vienna Scientific Cluster (VSC), project number 71878, for a demonstration of a Lagrangian re-analysis. We also acknowledge the use of ECMWF's computing and archive facilities, provided through a special project (spatvojt). This study was supported in part by the European Union's Horizon Europe research and innovation program (EYE-CLIMA) under grant agreement number 101081395, by the Edu4ClimAte program under grant agreement number 101071247, and by NOAA cooperative agreement NA22OAR4320151, for the Cooperative Institute for Earth System Research and Data Science (CIESRDS).

We thank Katharina Meixner und Andreas Engel (Goethe University Frankfurt) for the interesting and fruitful discussions. Finally, we thank Marina Dütsch, Lucie Bakels, Silvia Bucci, Katharina Baier, Daria Tatsii, Anjumol Raju, Rakesh Subramanian, Sophie Wittig, Omid Nabavi, Ioanna Evangelou, Michael Blaschek, Andrey Skorokhod, Benjamin Püschel, Luise Kandler, and Petra Seibert (University of Vienna) for their support.

# Appendix A: Evaluation of inversion setup sensitivities


This Appendix presents the results from various inversion setups, organized by specific aspects of the inversion process to examine the sensitivity of results to each setting. We display emission time series for four major emitting countries: the UK, Germany, France, and Italy, as well as for the aggregated EU-27 emissions. Where relevant, we include a priori or a posteriori emission maps, as well as inversion increments (a posteriori - a priori), either averaged over the entire study period or focused on specific years or countries. Additionally, we provide error reduction information for one particular case to provide further insight. The reference inversion is indicated by a black frame around the respective inversion maps or by a thick line in the case of emission time series.

## **Appendix B: Sensitivity to the a priori emissions**








We employ seven different *a priori* emission fields derived from the GAINS inventories, UNFCCC reports, and EDGAR data (see Sect. 2.3 and Fig. 3). Figure B1 shows the inversion increments averaged throughout the entire study period when using different a priori inventories. While in some regions the increments vary in magnitude, they show a very similar pattern across all cases. We see negative increments in northern Germany, especially in the area around Cologne, where a priori emissions tend to be very high. The UNFCCC- and EDGAR-based inversions also show strong negative increments in Berlin, where the a priori estimates are high. All tests show large positive increments in southwest Germany, positive increases in France, Italy, and the UK, and negative increments in Switzerland. Notice that the EDGAR-based inventories E7P and E8 show negative increments in the area of Moscow, in contrast to the other inventories. For a more detailed analysis, Fig. B2 presents the a priori (a–g) and a posteriori (h–n) emissions in the Moscow region, averaged over the study period 2005-2021. While the a priori emissions exhibit significant differences, the a posteriori emissions show better agreement. However, substantial uncertainties persist due to the use of different a priori inventories, even after the correction from the inversion.

Figure B3 presents the a posteriori emission time series for the UK, Germany, France, Italy, and the EU-27 based on different a priori emission inventories. In the UK (Fig. B3a), differences in the a posteriori emissions due to different a priori inventories are relatively large in the early years of the study period, but decrease significantly toward the end, particularly after 2011, when the British observation network was expanded. Similarly, the differences in the French a posteriori SF<sub>6</sub> emissions (Fig. B3c) decrease over the study period; however, this is less evident than for the UK. In Germany (Fig. B3b), the sensitivity to different inventories remains relatively low throughout the study period. This is particularly noteworthy toward the end of the period, when a priori inventories show considerable differences, indicating substantial improvements from the optimization. In Italy (Fig. B3d), a priori emissions from the different inventories are quite similar until 2017, leading to relatively closely aligned a posteriori emissions. However, toward the end of the time series, the a priori estimates start to diverge, resulting in larger differences in the corresponding a posteriori emissions, especially in 2021. This divergence is likely related to the fact that Monte Cimone provided observations only until 2017, after which constraints on Italian emissions were substantially reduced. A closer examination of the year 2021 reveals significantly higher a priori emissions and positive increments close to Milan for the EDGAR- and UNFCCC-based inventories (Fig. B4c-g/j-n), compared to the lower values from the GAINS-based inventory (Fig. B4a,b/h,i). The larger increments observed for higher a priori values can most likely be attributed to the definition of the a priori emission uncertainty, giving the algorithm more freedom in grid cells with high a priori values. Consequently, emissions that are high but still underestimated are easier to correct than those that are even lower. This might indicate that our uncertainties for individual grid cells might be generally too small and that our posterior uncertainties, as estimated using the analytical method, might be significantly underestimated at the national level. The aggregated a posteriori emissions of the EU-27 countries (Fig. B3e) show a relatively low dependence on the a priori inventory, with differences across cases decreasing up to 2018. After 2018 differences slightly grow again, which can be attributed to the strongly diverging a priori inventories.

**Figure B1.** Inversions Inversion increments (a posteriori - a priori) averaged over the entire study period (2005-2021), shown for different a priori emission inventories: (a) GS, (b) GS\_HR, (c) UP, (d) UN, (e) E7P, (f) E7N, (g) E8.

**Figure B2.** Moscow region: a priori emissions (left) and a posteriori emissions averaged over the entire study period (2005-2021) using different inventories: (a)/(h) GS, (b)/(i) GS\_HR, (c)/(j) UP, (d)/(k) UN, (e)/(l) E7P, (f)/(m) E7N, (g)/(n) E8.

**Figure B3.** Annual emission time series for (a) the United Kingdom, (b) Germany, (c) France, (d) Italy, and (e) the EU-27, using different a priori emissions inventories. The colored solid lines (light blue: GS, dark blue: GS\_HR, light green: UP, dark green: UN, light red: E7P, dark red: E7N, orange: E8) represent the a posteriori emissions derived using different a priori emission inventories, which are shown by the dashed lines in corresponding colors. The vertical grey-gray lines indicate the times when additional observational data from the expansion of the UK network became available.

**Figure B4.** A priori emissions (left-hand side) and emission increments (right-hand side) in Italy for the year 2021, shown for different a priori emission inventories: (a)/(h) GS, (b)/(i) GS\_HR, (c)/(j) UP, (d)/(k) UN, (e)/(l) E7P, (f)/(m) E7N, (g)/(n) E8.

## Appendix C: Sensitivity to the a priori emissions emission uncertainty

In FLEXINVERT+, the *a priori* emission uncertainty in each grid cell is calculated as a fraction of the corresponding emission value. We evaluate four different settings with fractions of 30%, 50%, 70%, and 100%, with the inversion results presented in Fig. C1. As the *a priori* emission uncertainty increases from 30% to 100%, the constraint on the *a priori* emissions weakens, allowing the *a posteriori* emissions to deviate further from their *a priori* values. Notable, the step from 30% to 50% results in the largest differences, while the step from 70% to 100% causes only minor differences. Using prior uncertainties of 30%, 50%, and 70% resulted in reduced chi-square values close to 1 (1.12, 0.87, and 0.70, respectively, averaged over all study years). By contrast, the value of 0.55 obtained for 100% uncertainty might indicate an overestimation of the a priori errors. In general, we observe little sensitivity to the *a priori* emissions in Germany (Fig. C1b) and the UK (Fig. C1a). For France, Italy, and the aggregated EU-27 emissions, the sensitivity to the *a priori* uncertainty generally decreases over time, with differences becoming relatively small after 2014.

Furthermore, we test various minimum emission uncertainty values to allow the algorithm to adjust emissions in grid cells with very small *a priori* values. The seven minimum values tested are  $5 \times 10^{-14}$ ,  $1 \times 10^{-13}$ ,  $5 \times 10^{-13}$ ,  $1 \times 10^{-12}$ ,  $5 \times 10^{-12}$ ,  $1 \times 10^{-11}$ , and  $5 \times 10^{-11}$  kg m<sup>-2</sup> h<sup>-1</sup>, with the corresponding uncertainty distribution illustrated in Figure C2. Figure C3 shows the respective inversion results for (a) the United Kingdom, (b) Germany, (c) France, (d) Italy, and (e) the EU-27. For the smaller minimum emission uncertainties between  $5 \times 10^{-14}$  and  $1 \times 10^{-12}$  kg m<sup>-2</sup> h<sup>-1</sup> (red, blue, green, and purple), the inversion results remain very similar. However, as the minimum emission uncertainty further increases, the *a posteriori* emissions deviate more significantly from the *a priori* values. Similar to the previous tests, the inversion results are very stable for the UK and Germany, while the sensitivity in France decreases notably from 2012 to 2021. In contrast, Italy's *a posteriori* emissions exhibit relatively large differences in some years (including after 2014), likely due to limited observational coverage in southern Italy. These differences are particularly evident for the highest tested value of  $5 \times 10^{-11}$  kg m<sup>-2</sup> h<sup>-1</sup> (brownlight orange). In case of the aggregated EU-27 *a posteriori* emissions, the highest tested value of  $5 \times 10^{-11}$  kg m<sup>-2</sup> h<sup>-1</sup> also leads to considerably higher interannual variability compared to the other tested values.

To further investigate how the minimum a priori uncertainty affects the inversion, we also show the 2012 European inversion increments for the same seven values in Fig. C4, since we observe the biggest differences in this year. Consistent with Fig. C3, increments are similar for minimum values between  $5 \times 10^{-14}$  and  $1 \times 10^{-12}$  kg m<sup>-2</sup> h<sup>-1</sup>. However, as the minimum a priori uncertainty further increases, the emission increment patterns begin to change. The increments become less localized and spread over larger areas, especially apparent in France. This shift occurs because the *a priori* uncertainty becomes similar across most grid cells (see Figure C2), compelling the algorithm to distribute increments more evenly. For poorly observed areas, such as regions around the Black Sea, the large a priori uncertainties result in excessively strong positive increments. This behavior contrasts with well-covered areas such as Germany or the UK, where almost no differences are observed until the minimum value reaches  $5 \times 10^{-11}$  kg m<sup>-2</sup> h<sup>-1</sup> (g), at which point the results become noisy in the whole domain, indicating that the inversion becomes poorly constrained. At this point the inversion problem becomes rather ill-posed, producing widespread artifacts.

**Figure C1.** Annual emission time series for (a) the United Kingdom, (b) Germany, (c) France, (d) Italy, and (e) the EU-27, using differnt settings of a priori emissions uncertainties, defined as a fraction of the corresponding emission value in each grid cell. The colored solid lines (red: 30%, orange: 50%, light blue: 70%, and blue: 100% of the respective emission value) represent the a posteriori emissions derived using the different settings and the gray dashed line shows the a priori emissions. The vertical grey gray lines indicate the times when additional observational data from the expansion of the UK network became available.

Figure C2. A priori emission uncertainties averaged over the entire study period (2005-2021), shown for different tested minimal a priori emission uncertainty values (a)  $5 \times 10^{-14} \,\mathrm{kg \ m^{-2} \ h^{-1}}$ , (b)  $1 \times 10^{-13} \,\mathrm{kg \ m^{-2} \ h^{-1}}$ , (c)  $5 \times 10^{-13} \,\mathrm{kg \ m^{-2} \ h^{-1}}$ , (d)  $1 \times 10^{-12} \,\mathrm{kg \ m^{-2} \ h^{-1}}$ , (e)  $5 \times 10^{-12} \,\mathrm{kg \ m^{-2} \ h^{-1}}$ , (f)  $1 \times 10^{-11} \,\mathrm{kg \ m^{-2} \ h^{-1}}$ , and (g)  $5 \times 10^{-11} \,\mathrm{kg \ m^{-2} \ h^{-1}}$ ).

Figure C3. Annual emission time series for (a) the United Kingdom, (b) Germany, (c) France, (d) Italy, and (e) the EU-27, testing different minimal a priori emission uncertainty values. The colored solid lines (dark blue:  $5 \times 10^{-14}$ , blue:  $1 \times 10^{-13}$ , green:  $5 \times 10^{-13}$ , purple:  $1 \times 10^{-12}$ , red:  $5 \times 10^{-12}$ , orange:  $1 \times 10^{-11}$ , and light orange:  $5 \times 10^{-11}$  kg m<sup>-2</sup> h<sup>-1</sup>) represent the a posteriori emissions derived using the different settings and the gray dashed line shows the a priori emissions.

Figure C4. Inversions Inversion increments (a posteriori - a priori) for the year 2012, shown for different tested minimal a priori emission uncertainty values (a)  $5 \times 10^{-14} \, \mathrm{kg \ m^{-2} \ h^{-1}}$ , (b)  $1 \times 10^{-13} \, \mathrm{kg \ m^{-2} \ h^{-1}}$ , (c)  $5 \times 10^{-13} \, \mathrm{kg \ m^{-2} \ h^{-1}}$ , (d)  $1 \times 10^{-12} \, \mathrm{kg \ m^{-2} \ h^{-1}}$ , (e)  $5 \times 10^{-12} \, \mathrm{kg \ m^{-2} \ h^{-1}}$ , (f)  $1 \times 10^{-11} \, \mathrm{kg \ m^{-2} \ h^{-1}}$ , and (g)  $5 \times 10^{-11} \, \mathrm{kg \ m^{-2} \ h^{-1}}$ ).

## Appendix D: Sensitivity to the spatial correlation of the a priori emission uncertainty





FLEXINVERT+ uses an exponential decay function to account for spatial emission uncertainty correlations. We test various spatial scale lengths of 50, 100, 250, 500, 1000 km, as well as a configuration without any spatial correlation. Figure D1 presents the emission time series for these tests. As the correlation length increases, the observational information influences a larger number of emission grid cells, causing the aggregated a posteriori emissions to deviate more from their a priori values. With high correlation lengths, the algorithm's ability to capture the spatial variability of the emissions is limited. Conversely, if the correlation length is very small, there will be insufficient observational constraint on the emissions. As a consequence, the number of grid cells with substantial inversion increments increases with growing correlation length (see Fig. D2, increments averaged over the study period). At the same time, the modeled error reduction substantially increases (see Fig. D3, error reduction shown for the year 2012). However, this is solely due to the much larger a priori uncertainties, and thus should not be interpreted as an indication for of a superior inversion quality. It reflects the broader spatial distribution of observational information rather than an actual improvement in the ability of the inversion to constrain emissions. Our findings align closely with Thompson et al. (2011), who observed a similar error reduction trend when testing correlation lengths between 50 and 2000 km in a European N<sub>2</sub>O inversion. Our tests further show that in the absence of spatial correlation, the inversions can yield substantial negative emissions at the grid-cell level (down to  $-21 \text{ pg m}^{-2} \text{ s}^{-1}$ ). Such values are unphysical and indicate that the problem is poorly constrained. In contrast, imposing large correlation lengths of 500 and 1000 km slightly reduces the agreement between observed and posterior mole fractions (see Table S9 and S10), as the imposed correlation limits the inversion's ability to resolve regional emission patterns. Across all investigated regions (Fig. D1), we observe that after 2012, inversion results become less sensitive to the choice of correlation length, especially in Germany (Fig. D1b) and the UK (Fig. D1a), where error reduction is highest (Fig. D3) and inversion results show remarkable stability. In France (Fig. D1c) and Italy (Fig. D1d), the error reduction is smaller and a posteriori emissions remain generally more sensitive to the correlation length. For the aggregated EU-27 emissions (Fig. D1e), results are highly sensitive to the chosen spatial correlation length, though they also stabilize after 2012. Notably, results for correlation lengths of 50, 100, and 250 km are relatively close, while values of 500 km, 1000 km, and no correlation show great deviations in case of the aggregated EU-27 emissions.

**Figure D1.** Annual emission time series for (a) the United Kingdom, (b) Germany, (c) France, (d) Italy, and (e) the EU-27, testing different spatial scale lengths for the a priori emission uncertainty correlation. The colored solid lines (blue: 50km, green: 100 km, purple: 250 km, orange: 500 km, light orange: 1000 km, and red: no correlation) represent the a posteriori emissions derived using the different settings and the gray dashed line shows the a priori emissions. The vertical grey gray lines indicate the times when additional observational data from the expansion of the UK network became available.

**Figure D2.** Inversions Inversion increments (a posteriori - a priori) averaged over the study period (2005-2021), shown for different spatial scale lengths for the a priori emission uncertainty correlation: (a) no correlation, (b) 50 km, (c) 100 km, (d) 250 km, (e) 500 km, and (f) 1000 km

**Figure D3.** Error reduction for the year 2012, shown for different spatial scale lengths for the a priori emission uncertainty correlation: (a) no correlation, (b) 50 km, (c) 100 km, (d) 250 km, (e) 500 km, and (f) 1000 km

#### 640 Appendix E: Sensitivity to the observation datasets






We perform tests using subsets of the global observation dataset: (1) the entire global dataset, (2) the global dataset excluding mountain stations, (3) the global dataset selecting night observations (00:00–06:00) at mountain stations and afternoon observations (12:00–18:00) at other sites, (4) the global dataset using exclusively high-frequency surface observations, (5) the European dataset, a subset of the global dataset including solely the observations in and around Europe (see Fig.;1(see Sect 2.1), (6) the European dataset excluding mountain stations, (7) the European dataset selecting night observations at mountain stations and afternoon observations at other sites, and (8) the European dataset using exclusively high-frequency surface observations. Figure E1 presents the emission time series using these datasets.

For the UK, Germany, France and Italy, the choice between the global and European datasets has a small impact on the inversion results; however, for the aggregated EU-27 emissions, differences can be pronounced. This indicates that distant observations barely constrain the emissions in relatively well-observed countries, but they still help to constrain emissions in less well-covered areas have little influence on emissions in countries that are well observed, but can affect emissions in areas that are less well covered. Excluding observations from mountain stations (Fig. E1, red lines) has a minimal impact on the UK emissions (Fig. E1a) and also shows a limited effect in France (Fig. E1c) after 2008. In Germany (Fig. E1b) and the EU-27 (Fig. E1e) the exclusion of mountain stations can lead to notable differences in certain years, while for Italian emissions (Fig. E1d), the impact can be substantial, such as in 2016. Figure E2 shows the 2016 inversion increments, illustrating how the exclusion of the mountain stations (Fig. E2b,f) such as JFJ, ZSF, and MCNCMN, leads to large positive increments in Switzerland and nearby areas (including North Italy), as the observational coverage of this region is drastically reduced. We assume that the limited observational coverage causes the region to be influenced by the high emission increments in southwestern Germany. Selecting only afternoon/night observations (Fig. E1, green lines) generally results in posterior emissions closer to the prior values due to reduced number of available observations. Similarly, the inversion increments (Fig. E2c,g) are attenuated, however, the patterns of emission increments remain rather similar. Excluding flask measurements and data from moving platforms affects early study years (e.g. emissions the UK in Fig. E1a, purple lines), but as the observational coverage with on-line measurement stations increases, the impact of these additional measurements becomes negligible. Similar to the other sensitivity tests, we observe that the sensitivity to the used datasets decreases over the study period, however, 2016 and 2017 stand out as exceptions, likely due to the exceptionally high emissions in southwestern Germany during these years.

Figure E1. Annual emission time series for (a) the United Kingdom, (b) Germany, (c) France, (d) Italy, and (e) the EU-27, testing different subsets of the observation dataset. The colored lines (solid blue: the full global dataset, solid red: the global dataset excluding mountain stations, solid green: the global dataset selecting night observations at mountain stations and afternoon observations at other sites, solid purple: the global dataset using exclusively high-frequency surface observations, dotted blue: the European dataset, a subset of the global dataset focused on observations in and around Europe, dotted red: the European dataset excluding mountain stations, dotted green: the European dataset selecting night observations at mountain stations and afternoon observations at other sites, dotted purple: the European dataset using exclusively high-frequency surface observations) represent the a posteriori emissions derived using the different settings and the gray dashed line shows the a priori emissions. The vertical grey gray lines indicate the times when additional observational data from the expansion of the UK network became available.

**Figure E2.** Inversions Inversion increments (a posteriori - a priori) averaged over for the study period (2005-2021) year 2016, shown for different observation datasets: (a) the full global dataset, (b) the global dataset excluding mountain stations, (c) the global dataset selecting night/afternoon observations, (d) the global dataset using exclusively high-frequency surface observations, (e) the European dataset, a subset of the global dataset focused on observations in and around Europe, (f) the European dataset excluding mountain stations, (g) the European dataset selecting night/afternoon observations (h) the European dataset using exclusively high-frequency surface observations.

### Appendix F: Sensitivity to the observation uncertainty




We explore multiple configurations for the observation uncertainty, testing constant values of 0.02, 0.04, 0.06, 0.08, and 0.1 ppt. Additionally, we use two approaches where we (i) base the observation error on the RMSE between a priori modeled and observed values, averaged by station, and (ii) estimate the model error from the standard deviation of the a posteriori error distribution at each station, using initial inversion runs. Figure F1 presents the emission time series of these tests. As observation uncertainty increases, the observational constraint weakens, causing the a posteriori emissions to follow more closely their a priori values. The two approaches that account for spatial variability in the model uncertainty generally fall within the range of constant-error settings, although they show a slightly different pattern for some periods. The inversion results show very low sensitivity to the observation uncertainty in the UK and Germany, especially after 2012 when results are extremely stable. For France, Italy, and the EU-27, the sensitivity to the observation uncertainty also declines after 2012. The reduced chi-squared values averaged over the study period were 3.65, 1.50, 0.87, 0.57, and 0.41 for assumed observation errors of 0.02, 0.04, 0.06, 0.08, and 0.10 ppt, respectively. These results suggest that an observation error of about 0.06 ppt provides the most consistent fit, while 0.02 ppt underestimates and 0.10 ppt overestimates the uncertainties. Note at this point that the smallest error setting of 0.02 ppt (red) - which is likely an underestimation of the actual uncertainty - shows the greatest deviation from the other tests. The chi-squared values related to the spatial varying uncertainties were 0.80 (standard deviation, a posteriori distribution), and 0.31 (RMSE, a priori distribution). Indeed, using the RMSE of the a priori distribution is expected to be an overestimation, as it also reflects systematic mismatches and biases in the a priori emissions rather than purely observational and transport model uncertainty.

**Figure F1.** Annual emission time series for (a) the United Kingdom, (b) Germany, (c) France, (d) Italy, and (e) the EU-27, testing various observation error settings. The colored solid lines (red: 0.02 ppt, orange: 0.04 ppt, light orange: 0.06 ppt, light blue: 0.08 ppt, blue: 0.1 ppt, light green: standard deviation of the a posteriori distribution, green: RMSE between *a priori* modeled and observed values) represent the a posteriori emissions derived using the different settings and the gray dashed line shows the a priori emissions. The vertical grey gray lines indicate the times when additional observational data from the expansion of the UK network became available.

#### 685 Appendix G: Sensitivity to the baseline optimization

FLEXINVERT+ includes an option for baseline optimization, where spatially aggregated contributions are adjusted on a coarse grid. We test different coarse grid resolutions by dividing the global field into 8, 4, and 2 latitude bands, with northern boundaries at  $[-60^{\circ}, -30^{\circ}, -15^{\circ}, 0^{\circ}, 15^{\circ}, 30^{\circ}, 60^{\circ}, 90^{\circ}]$ ,  $[-30^{\circ}, 0^{\circ}, 30^{\circ}, 90^{\circ}]$ , and  $[0^{\circ}, 90^{\circ}]$ , respectively. In addition, we run inversions where the entire global field is optimized with a single scalar. Figure G1 presents the emission time series for these tests. For all regions studied, we find that optimizing the baseline using 2, 4, or 8 latitudinal bands has minimal impact on the results. However, optimizing the entire field in a single global grid cell results in significantly higher a posteriori emissions, particularly before 2012, especially evident for the EU-27 emissions (Fig. G1e). This trend is also evident in the inversion increments (Fig. G2), where the positive increments are larger, and the negative increments are less pronounced when optimizing the whole field (Fig. G2a). These results can be linked to the large inter-hemispheric gradient in atmospheric SF<sub>6</sub> mole fractions. Potential biases in the modeled SF<sub>6</sub> mole fraction fields likely differ between the Southern and Northern Hemispheres, making a single optimization factor insufficient to represent both regions accurately. However, as the observational coverage increases, the sensitivity to the spatial resolution of the baseline drastically decreases and inversion results become extremely stable for all tested regions.

We also test different temporal baseline optimization intervals of 15, 30, 45, and 60 days, with inversion results shown in Fig. G3. The a posteriori emissions are only minimally sensitive to the choice of the temporal interval between 15 and 60 days. Although small differences occasionally appear in certain years and regions, the overall inversion results remain highly stable. Furthermore, we test various baseline uncertainty values set to 0.0001, 0.0003, 0.0005, 0.0007, 0.0009, 0.001, 0.01, 0.1, and 1 ppt and run an inversion without any baseline optimization. The resulting a posteriori emission time series are shown in Fig. G4. The baseline optimization consistently reduces the a posteriori emissions across all regions, indicating that the optimization tends to shift the a posteriori baseline to higher values. At a baseline uncertainty of 0.0001 ppt, the changes in a posteriori emissions are minimal. However, increasing the uncertainty to 0.0003 ppt produces a notable decrease in emissions. Further increases in the uncertainty continue to lower the a posteriori emissions, though the effect diminishes with each step, converging toward stable results between 0.01 and 0.1 ppt. Increasing the baseline uncertainty up to 0.01 ppt also improves the bias between a posteriori modeled and observed mole fractions while higher uncertainties yield the same results (see Table S11-S13). Figure G5 presents the inversion increments for uncertainty values between 0.0001 ppt and 0.01 ppt, as well as for the case without optimization. Consistent with Fig. G4, we observe a decrease in increments with increasing baseline uncertainty. As observed in other sensitivity studies, the sensitivity to baseline optimization decreases significantly over the course of the study period, with results stabilizing toward the end for all tested regions.

**Figure G1.** Annual emission time series for (a) the United Kingdom, (b) Germany, (c) France, (d) Italy, and (e) the EU-27, testing various spatial resolutions of aggregated baseline contributions for the baseline optimization. The colored solid lines represent the a posteriori emissions derived when optimizing the baseline, regarding the different spatial resolutions (red: the whole global field, orange: 2 latitudunal bands, light blue: 4 latitudunal bands, and blue: 8 latitudunal bands). The gray dashed line shows the a priori emissions. The vertical grey gray lines indicate the times when additional observational data from the expansion of the UK network became available.

**Figure G2.** Inversions Inversion increments (a posteriori - a priori) averaged over the study period (2005-2021), shown for various spatial resolutions of aggregated baseline contributions for the baseline optimization: (a) the whole field, (b) 2 latitudinal bands, (c) 4 latitudinal bands, and (d) 8 latitudinal bands

**Figure G3.** Annual emission time series for (a) the United Kingdom, (b) Germany, (c) France, (d) Italy, and (e) the EU-27, testing different temporal baseline optimization intervals. The colored solid lines (pink: 15 days, orange: 30 days, light green: 45 days, and green: 60 days) represent the a posteriori emissions. The gray dashed line shows the a priori emissions. The vertical grey gray lines indicate the times when additional observational data from the expansion of the UK network became available.

**Figure G4.** Annual emission time series for (a) the United Kingdom, (b) Germany, (c) France, (d) Italy, and (e) the EU-27, testing baseline uncertainty values. The colored solid lines represent the a posteriori emissions (dark red: no optimization, red: 0.0001 ppt, dark orange: 0.0003 ppt, orange: 0.0005 ppt, light orange: 0.0007 ppt, light green 0.0009 ppt, green 0.001 ppt, dark green 0.01 ppt, light blue 0.1 ppt, and blue: 1 ppt). These tests refer to a baseline optimization using 4 latitudinal bands and a 30-day temporal time window. The gray dashed line shows the a priori emissions. The vertical grey-gray lines indicate the times when additional observational data from the expansion of the UK network became available.

**Figure G5.** Inversions Inversion increments (a posteriori - a priori) averaged over the study period (2005-2021), shown for various baseline uncertainty values: (a) no optimization, (b) 0.0001 ppt (c) 0.0003 ppt, and (d) 0.0005 ppt, (e) 0.0007 ppt, (f) 0.0009 ppt, (g) 0.001 ppt, (h) 0.01 ppt

Figure H1. European inversion grid with variable cell sizes, featuring configurations of (a) 7.229 and (b) 588 grid cells

### Appendix H: Sensitivity to the emission grid

We utilize emission grids with varying cell sizes, created by aggregating cells with low emission contributions based on emission sensitivities and *a priori* emissions (see details in Thompson and Stohl, 2014). The tested grids include configurations with 588, 744, 1,992, 2,781, 4,248, 5,370, and 7,229 grid cells, each remaining constant over time. Additionally, we explore three dynamic setups where the grid configuration adjusts annually, with the number of cells ranging from (i) 2,781 to 5,916, (ii) 3,645 to 6,599, and (iii) 4,151 to 7,229. Figure H1 illustrates the emission grid with (a) the highest and (b) the lowest number of grid cells, while Fig. H2 displays the inversion results for all tested grid configurations. The inversion results demonstrate minimal sensitivity to the number of grid cells within the tested range, with only minor differences observed in isolated years and regions.

**Figure H2.** Annual emission time series for (a) the United Kingdom, (b) Germany, (c) France, (d) Italy, and (e) the EU-27, testing various emission grid configurations. The colored solid lines represent the a posteriori emissions (dark red: 588, red: 744 ppt, dark orange: 1,992, orange: 2,781, light orange: 4,248, light blue 5,370, blue 7229, light green: 2,781 - 5,916, green 3,645 - 6,599, and dark green: 4,151-7,229 grid cells). The gray dashed line shows the a priori emissions. The vertical grey gray lines indicate the times when additional observational data from the expansion of the UK network became available.

# Appendix I: Sensitivity to the whole inversion ensemble

Fig. I1 shows the results of all performed inversions, displaying the full set of a posteriori emissions alongside the average across all sensitivity tests. Our results show that sensitivity to the various inversion settings decreases significantly after 2012, aligning with the expansion of the British observation network. This trend is particularly evident in the UK and Germany, where the results become highly stable across all sensitivity tests. Even under unfavorable settings that lead to outliers during periods of limited observational coverage, results remain stable towards the end of the study period.

Figure I1. Annual emission time series for (a) the United Kingdom, (b) Germany, (c) France, (d) Italy, and (e) the EU-27. The eolored solid gray lines represent the inversion results of all sensitivity tests and the solid black lines represent the average a posteriori emissions across all performed tests. The vertical gray lines indicate the times when additional observational data from the expansion of the UK network became available. The vertical gray lines indicate the times when additional observational data from the expansion of the UK network became available.

## **Appendix J: Selection of parameter ranges**


- A posteriori emission values for the United Kingdom, Germany, France, Italy, and the EU-27 for the period 2005 to 2021. The values present the average over the entire Monte Carlo inversion ensemble with 2-σ uncertainty ranges. year United Kingdom tyr<sup>-1</sup>Germany tyr<sup>-1</sup>France tyr<sup>-1</sup>Italy tyr<sup>-1</sup>EU-27 tyr<sup>-1</sup>2005 41 ± 13 166 ± 41 88 ± 37 31 ± 12 484 ± 2132006 60 ± 11 157 ± 15 70 ± 30 36 ± 17 350 ± 692007 46 ± 18 162 ± 35 58 ± 47 43 ± 36 342 ± 882008 65 ± 13 121 ± 57 112 ± 49 46 ± 17 370 ± 1192009 47 ± 16 135 ± 28 48 ± 24 54 ± 25 358 ± 1052010 42 ± 13 119 ± 20 58 ± 25 67 ± 33 368 ± 972011 37 ± 29 99 ± 39 59 ± 32 42 ± 13 308 ± 692012 53 ± 10 112 ± 26 93 ± 42 65 ± 44 363 ± 872013 31 ± 11 91 ± 28 98 ± 36 59 ± 32 332 ± 952014 37 ± 9 142 ± 17 83 ± 34 33 ± 7 364 ± 662015 33 ± 8 137 ± 13 65 ± 22 55 ± 33 351 ± 962016 33 ± 6 153 ± 38 94 ± 19 49 ± 26 390 ± 942017 27 ± 6 205 ± 42 67 ± 27 64 ± 48 469 ± 1442018 20 ± 6 105 ± 15 61 ± 21 39 ± 15 291 ± 622019 17 ± 6 118 ± 11 64 ± 33 42 ± 21 308 ± 752020 18 ± 5 118 ± 22 64 ± 17 35 ± 19 357 ± 942021 20 ± 6 95 ± 11 51 ± 28 37 ± 24 255 ± 58Based on the insights from our sensitivity tests, we defined the final parameter ranges for the inversion ensemble. Compared to the broader ranges explored in the sensitivity tests, these final ranges are narrowed to exclude (i) unlikely values that could lead to extreme or problematic results and (ii) parameters to which the inversion showed negligible sensitivity.
  - Inversion settings generated using Monte Carlo methods. Parameters are sampled either continuously from a uniform distribution within specified ranges or discretely from predefined values, as indicated in the table header. All inversions employ an emission grid configuration of 558 cells, and the baseline is optimized in 8 latitudinal bands with a temporal window of 30 days and a baseline uncertainty of 0.1 ppt. Identifier **A priori emission uncertainty**(
    - (i) A priori emission uncertainty: We adopt a normal distribution with mean 0.5 -1.0) Minimum a priori emission uncertainty(1.0e-13-1.0e-11) kgm<sup>-2</sup>h<sup>-1</sup> and standard deviation 0.1. Within this range, the inversion yields reasonable results and reduced chi-squared values close to one (see Appendix C).
- (ii) Minimal a priori emission value: We adopt a normal distribution with mean 5 × 10<sup>-13</sup>kg m<sup>-2</sup> h<sup>-1</sup> and standard deviation 1 × 10<sup>-13</sup>kg m<sup>-2</sup> h<sup>-1</sup>. Values above this range lead to less localized inversion increments that spread over larger areas. This could potentially introduce biases in regions with sparse observational constraints (see Appendix C).
  - (iii) Spatial correlation of the a priori emission uncertainty: (50-250) We adopt a normal distribution with mean 250 km and standard deviation 100 km. This choice represents our best estimate of a compromise between sufficiently constraining emissions and maintaining the inversion's ability to resolve regional emission patterns (see Appendix D).
  - (iv) Observation uncertainty: (0.03-0.1)A priori emission inventoriesall 7 inventories (see Sect. 2.3)Observation datasetall 8 datasets (see Sect. 2.5) 0 0.94 8.8e-12 207 0.051 UN Europe: night/afternoon selection 1 0.63 1.1e-12 108 0.035 E7N Global 2 0.84 7.3e-12 55 0.045 E7N Europe: night/afternoon selection 3 0.96 5.7e-12 230 0.044 GS\_HR Global: high-frequency surface stations 4 0.64 9.2e-12 244 0.030 GS\_HR Global 5 0.96 8.7e-12 110 0.051 E7N Global 6 0.71 9.3e-12 158 0.043 GS\_HR Europe 7 0.87 2.3e-12 123 0.080 E7P Europe: night/afternoon selection 8 0.81 1.9e-12 160 0.071 UP Global: night/afternoon selection 9 0.51 1.5e-12 136 0.047 E7N Global 10 0.65 6.9e-12 55 0.065 E7P

Global: high-frequency surface stations 11 0.83 6.3e-12 78 0.066 E7P Global: night/afternoon selection 12 0.76 3.1e-12 128 0.047 UN Global: excluding mountain stations 13 0.71 4.3e-12 220 0.072 E7N Europe: high-frequency surface stations 14 0.55 3.4e-12 98 0.054 E7P Global 15 0.67 6.3e-12 116 0.037 UP Europe: high-frequency surface stations 16 0.54 3.0e-12 63 0.078 GAINS Europe: night/afternoon selection 17 0.77 3.5e-12 68 0.072 UP Europe: night/afternoon selection 18 0.81 9.8e-12 237 0.061 E8 Global 19 0.70 5.5e-12 87 0.076 E8 Europe: excluding mountain stations 20 0.93 9.9e-12 131 0.047 GS\_HR Global: high-frequency surface stations 21 0.73 1.2e-12 100 0.042 GAINS Global: excluding mountain stations 22 0.99 1.4e-12 62 0.055 GAINS Global: excluding mountain stations 23 0.86 6.7e-12 221 0.077 E7N Europe: high-frequency surface stations 24 0.76 9.1e-12 221 0.047 E7P Europe: excluding mountain stations 25 0.97 1.6e-13 232 0.064 E8 Global: night/afternoon selection 26 0.79 3.2e-12 231 0.062 GS\_HR Europe: high-frequency surface stations 27 0.82 2.9e-12 78 0.041 E8 Europe: excluding mountain stations 28 0.51 8.3e-12 75 0.065 E7P Europe 29 0.75 3.0e-12 228 0.069 UN Global: high-frequency surface stations We adopt a normal distribution with mean 0.06 ppt and standard deviation 0.01 ppt. Uncertainties within this range yield stable inversion results, while reduced chi-squared values remain close to one (see Appendix F).








- (v) Baseline optimization: We optimize the baseline in eight latitudinal bands, which improves performance compared to using a single global factor. The baseline uncertainty is set to 0.1 ppt to avoid underestimation, as smaller values (especially < 0.01 ppt) can introduce biases. Inversion results were largely insensitive to the choice of baseline optimization time window, for which we therefore adopted a value of 30 0.77 1.1e-13 225 0.078 GAINS Global: high-frequency surface stations 31 0.74 3.4c-12 118 0.032 E8 Europe: night/afternoon selection 32 0.60 8.7c-12 232 0.064 GAINS Europe: high-frequency surface stations 33 0.83 4.1e-12 100 0.067 E7N Europe: excluding mountain stations 34 0.97 7.3e-12 237 0.075 E8 Europe 35 0.56 9.2e-12 169 0.045 GS\_HR Europe: high-frequency surface stations 36 0.90 3.0c-12 154 0.074 E7P Europe: night/afternoon selection 37 0.78 3.6c-12 98 0.041 E7P Global 38 0.64 6.8e-13 103 0.059 GS HR Global: excluding mountain stations 39 0.71 5.6e-12 145 0.054 UP Europe: night/afternoon selection 40 0.56 9.8c-12 219 0.074 E8 Global: excluding mountain stations 41 0.79 6.2c-12 210 0.031 E7P Europe: excluding mountain stations 42 0.58 8.0e-12 72 0.076 GAINS Global: excluding mountain stations 43 0.52 1.0e-12 206 0.070 E7P Global 44 0.93 4.7e-12 64 0.045 E8 Global: excluding mountain stations 45 0.75 9.0e-12 100 0.049 GAINS Europe: night/afternoon selection 46 0.81 7.8e-12 124 0.079 GAINS Europe 47 0.54 8.7e-12 52 0.062 E7N Europe: high-frequency surface stations 48 0.79 2.0e-13 148 0.050 E7P Europe: high-frequency surface stations 49 0.56 1.9e-12 160 0.065 E7N Europe: high-frequency surface stations 50 0.75 4.4e-12 134 0.037 UN Global: high-frequency surface stations 51 0.57 5.6e-12 197 0.035 E7P Europe: excluding mountain stations 52 0.76 8.3e-12 75 0.059 UN Global: high-frequency surface stations 53 0.58 4.7e-12 82 0.072 E8 Europe: high-frequency surface stations 54 0.87 5.8e-12 55 0.067 GAINS Global: high-frequency surface stations 55 0.62 3.0e-12 221 0.044 GAINS Global: night/afternoon selection 56 0.88 7.4e-12 76 0.045 UP Europe: high-frequency surface stations 57 0.52 7.2e-13 232 0.049 GAINS Europe: night/afternoon selection 58 0.81 2.9e-12 96 0.057 UP Global: night/afternoon selection days (see Appendix G).

- (vi) Emission grid: We employ an emission grid of 558 cells, since the sensitivity tests indicated negligible dependence of the results on the grid within the tested configurations (see Appendix H).
  - Ensemble uncertainty, presented as the  $2\sigma$  uncertainty across all a posteriori emissions of the Monte Carlo ensemble at each grid cell
- (vii) A priori and observational datasets: For these inputs, we sample randomly from the datasets described in Sect. 2.3 and Sect. 2.1, as there were no objective reasons to restrict their parameter space.

Annual emission time series for (a) the United Kingdom, (b) Germany, (c) France, (d) Italy, and (e) the EU-27. The colored lines represent all inversion results using the Monte Carlo-based settings. The solid black lines represent the average a posteriori emissions across all performed inversions,

Annual emission time series for northwest Europe. The solid black lines represent the average a posteriori emissions across all performed inversions, with with a 2-σ uncertainty range for each year for each year. The blue and black squares represent the results from Manning et al. (2022) using the InTEM model with inversion time frames set to 3- and 1-months, respectively

|     | . The statements, findings, conclusions, and recommendations are those of the author(s) and do not necessarily reflect the views of NOAA |
|-----|------------------------------------------------------------------------------------------------------------------------------------------|
| 810 | or the U.S. Department of Commerce.                                                                                                      |
|     |                                                                                                                                          |
|     |                                                                                                                                          |
|     |                                                                                                                                          |
|     |                                                                                                                                          |
|     |                                                                                                                                          |
|     |                                                                                                                                          |
|     |                                                                                                                                          |
|     |                                                                                                                                          |
|     |                                                                                                                                          |
|     |                                                                                                                                          |
|     |                                                                                                                                          |
|     |                                                                                                                                          |
|     |                                                                                                                                          |
|     |                                                                                                                                          |
|     |                                                                                                                                          |
|     |                                                                                                                                          |
|     |                                                                                                                                          |
|     |                                                                                                                                          |
|     |                                                                                                                                          |
|     |                                                                                                                                          |
|     |                                                                                                                                          |
|     |                                                                                                                                          |
|     |                                                                                                                                          |
|     |                                                                                                                                          |
|     |                                                                                                                                          |
|     |                                                                                                                                          |
|     |                                                                                                                                          |
|     |                                                                                                                                          |
|     |                                                                                                                                          |
|     |                                                                                                                                          |
|     |                                                                                                                                          |
|     |                                                                                                                                          |
|     |                                                                                                                                          |
|     |                                                                                                                                          |
|     |                                                                                                                                          |
|     |                                                                                                                                          |

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
