# Peer review of "A thousand inversions to determine Quantifying European SF6 emissions from 2005 to 2021 using a large inversion ensemble"

_EGUsphere, 2025_

## Author Comment (AC1)

**Response**

**Response to reviewer 1**

Review of "**A thousand inversions to determine European SF6 emissions from 2005 to 2021**" by Vojta et al. for publication in Atmospheric Chemistry and Physics

Vojta et al. present a comprehensive atmospheric inversion study estimating European SF6 emissions from 2005 to 2021. This study utilised a large number of measurement stations and an extensive combination of inversions to report a robust estimate of the country-level $SF_6$ emissions and their uncertainties. The study reports a declining trend in the SF6 emissions for major European countries attributed to the various EU regulations that came into effect during the study period. The manuscript is very well-written and structured.

Studies on European SF6 are scarce, making this manuscript a valuable contribution to a topic of significant scientific interest. The authors have done impressive work, and I strongly recommend its publication.

We would like to thank reviewer #1 for the constructive review of our manuscript. All comments and questions were very helpful and we have made the corresponding revisions in the final version of the manuscript.

In the response we use 2 different colors. The blue-colored text is the general answer to the reviewer's comments, while new (or rewritten) text in the manuscript is colored in green.

I have a few minor comments and questions listed below:

The authors employ a Gaussian prior error distribution within an analytical inversion framework. Even though the posterior emissions are reported to be consistently higher than the prior, were there any instances where the inversion produced negative posterior emissions at the grid-cell level? If so, how was this issue dealt with?

Yes, when employing a Gaussian error distribution, negative emissions can appear at the grid-cell level. In our sensitivity tests, this effect was particularly pronounced under two conditions: (i) when observational errors were set unrealistically low, leading to overfitting of the measurements, and (ii) when the spatial correlations in the a priori uncertainties were weak, leaving the system underconstrained. In both cases, the inversion has too much freedom to adjust individual grid cells without sufficient regularization, which can result in negative values even though true emissions are non-negative.

We added to Appendix D:

Our tests further show that in the absence of spatial correlation, the inversions can yield substantial negative emissions at the grid-cell level (down to $-21$ pg m$^{-2}$ s$^{-1}$). Such values are unphysical and indicate that the problem is poorly constrained.

We dealt with this problem by applying an inequality constraint on the a posteriori emissions, using the truncated Gaussian approach by Thacker (2007).

We added:

> For $SF_6$, positive fluxes are expected over land, but the inversion may still yield negative a posteriori values in some grid cells. To correct this, we apply the truncated Gaussian method of (Thacker, 2007), which enforces non-negativity as an inequality constraint. The adjusted fluxes $\hat{x}'$ are calculated as
>
> $$\hat{x}' = \hat{x} + \hat{B}P^T \left(P\hat{B}P^T\right)^{-1} (c - P\hat{x}), \tag{5}$$
>
> where $\hat{x}$ is the original a posteriori estimate, $\hat{B}$ the a posteriori error covariance matrix, $P$ the operator identifying violations, and $c$ the constraint vector.

Could the authors clarify how the averaged posterior uncertainty shown in Fig. 4d was computed? Specifically, was temporal correlation across different years taken into account in this calculation? Since the posterior uncertainties in Fig. 4d appear visually comparable in magnitude to the emissions themselves, despite Fig. 4b indicating substantial uncertainty reduction and the prior uncertainty being set at 50%.

Thank you very much for pointing that out. Indeed, there was a mistake regarding the colorbar in 4d which we corrected now:

[Figure]

The authors can consider making the y-axis range the same for Figs. 7b and 7c to highlight the different trends in the German emission time series across the two regions.

Yes, thank you. We made the y-axis range consistent!

**Response to reviewer 2**

The manuscript by Martin Vojta and co-authors does a thorough analysis of European SF6 emissions derived by inverse modelling. The study builds on a previous publication by the same authors but focuses on national scale emissions in Europe. The authors carefully re-explore sources of uncertainty in their inversion estimate and provide an updated uncertainty estimate through the application of a large (59 members) ensemble of inversions. Parameters that should be varied for these inversions were selected by first identifying those with largest impact on posterior results and finally ensemble members were selected by randomly scanning the parameter space through an ensemble of inversions. The general idea of the publication of improving uncertainty estimates of inverse modelling through sets of sensitivity inversions is not new. However, the systematic exploration of the parameter space and the large number of inversions to derive final posteriori emission and their uncertainties is novel and a way forward in the field. The applied methods are appropriate and state of the art. The paper is clearly structured and well written. The length of the manuscript with various appendices is somewhat discouraging and the publication would benefit from a certain degree of shortening. Furthermore, I have several general and specific comments that should be addressed before publication.

We would like to thank reviewer #2 for the valuable and constructive review of our manuscript. The suggestions for improvements were very helpful, and we incorporated almost all of them into the final version of the manuscript. As suggested by the reviewer, we also shortened the manuscript by moving some content into the supplementary material, where we also provide additional information.

In the response we use 2 different colors. The blue-colored text is the general answer to the reviewer's comments, while new (or rewritten) text in the manuscript is colored in green.

General comments

Title: Unfortunately, the title is misleading and needs to be amended. Just because an ensemble of inversions with 59 members was run over 17 years in yearly batches, we should not call this "a thousand inversions". The publication would lose none of its importance if the title would state correctly what was evaluated: a large ensemble of inversions.

We changed the title to:

Quantifying European $SF_6$ emissions from 2005 to 2021 using a large inversion ensemble

Selection of parameters and their ranges for ensemble construction: There are several questions connected to the construction of the inversion ensemble. First of all, there is very little quantitative information why certain parameters were selected while others were omitted from the ensemble. For example, L269/270 states that the inversions were 'most sensitive' to the 'baseline uncertainty'. Nevertheless, in L240 it is stated that baseline uncertainty was fixed for all ensemble members. Sec. 2.6 also states that parameter ranges were narrowed as compared to the sensitivity tests but why remains unclear. Second, the choice of uniform distributions for the parameters may also introduce more 'extreme' events than may be representative of the true uncertainty of the parameter space. The uniform distribution also makes the selection of the parameter range much more critical as compared to using a normal distribution. Some additional explanation of the choices needs to be given to clarify if these are reasonable choices to represent real uncertainty. My concern is that by using (or by over-representing) parameter values that are unlikely or even unreasonable the posterior ensemble uncertainty gets blown up. Third, what tests were done to assure that the ensemble is representative?

How different would the results look if another 59-member ensemble would have been selected or the number of ensemble members halved/doubled? How was the Monte Carlo sampling of the parameter space performed? Independent for each parameter and sample?

Thank you for pointing this out. We fully agree that the selection of parameters and their ranges for ensemble construction is a critical aspect of our study, and that our initial description required more detail. We also agree that the choice of a uniform distribution makes the selection of the parameter range much more critical, and that unreasonable parameter choices would likely blow up the a posteriori ensemble uncertainty and should be omitted. To address this concern, we, therefore, decided to recalculate our inversions using a normal distribution instead of a uniform distribution in the parameter space. We base the specific choices for parameter means and variances on the outcomes of our sensitivity tests, and discuss them in more detail in the revised manuscript. Therefore, the sensitivity analysis was slightly extended to refine and better explain the selection of parameter ranges. In addition, to assess the representativeness and robustness of our ensemble, we performed further tests as suggested by the reviewer: (i) constructing another 59-member ensemble, (ii) halving, and (iii) doubling the ensemble size. These additional experiments are now included and discussed in the manuscript.

Overall, the new results show similar trends, however, smaller a posteriori uncertainties (as expected by the reviewer) compared to using a uniform distribution. This likely led to an overestimation of the uncertainty in the previous version of the manuscript, at least if errors are indeed Gaussian distributed. In some regions, the magnitude of our results also changes moderately. We mainly attribute this to the narrower normally distributed uncertainty ranges and the final choice of parameterization for the spatial prior correlation length, where we now use a normal distribution (N(250km, 100km)) instead of a uniform distribution between 50 and 250km. This adjustment was primarily motivated by the observation that small correlation lengths, which were allowed under the original uniform distribution, could lead to unrealistically large negative values on a grid cell level (see also the related comment of Reviewer 1).

Yes, we rewrote and added:

[revised manuscript text omitted]

Validation: Although, the results presented in this study are exhaustive, I am missing a basic evaluation of model performance as expressed through comparison with observed concentration time series. This could be presented in a very condensed form through a Taylor plot with prior and posterior results or through a table giving statistics for the individual sites. It would help to better understand which observations are well represented by the model and which are less well captured and, hence, do not constrain emissions as much. Furthermore, and because the network was not stable over the study period, it would be good to get an overview which and how many observations were available each year (again this could be given as a plot or as a table).

Yes, we agree with the reviewer. We have added a Taylor plot showing both prior and posterior results, and we present additional statistics in a table for the individual sites. Furthermore, we provide statistics for specific sensitivity tests to refine and clarify the selection of parameter ranges. In addition, we include more information on the observation dataset, show the availability of observations, and provide an overview of the global dataset.

We rewrote to:

Figure S1 provides an overview of all the ground-based measurements globally, while Fig. 1 shows the stations in Europe.

To determine the influence of data selection criteria on our results, we created eight different subsets. 1) We used the entire global dataset (presented in Vojta et al., 2024), and 2) we created a European subset by excluding on-line stations outside Europe (BRW, CGO, COI, GSN, HAT, IZO, MLO, NWR, RPB, SMO, SPO, SUM, THD; see Fig.S1) while retaining the European sites[1] (BIK, BRM, BSD, CMN, HFD, JFJ, MHD, RGL, TAC, ZEP, and ZSF; see Fig. 1). Note that the stations SUM in Greenland and IZO in Tenerife are geographically closest to the European inversion domain. For these global and European datasets, we further refined the selection by: a) retaining only night observations (00:00 - 06:00) at mountain stations and afternoon observations (12:00 - 18:00) at all other sites for continuous monitoring stations; b) creating a data subset that excludes mountain stations, and c) generating a subset that omits low-frequency measurements and data from moving platforms, retaining only high-frequency surface observations. Table S1 provides the number of observations used from each dataset for each year, whereas Tab. S2 shows the availability of online measurements within and outside Europe

[1]We initially still kept the globally distributed flask measurements and observations from moving platforms to improve the baseline optimization

and added:

Table S7 and Fig.S3 demonstrate the statistical improvements at all continuous surface stations, with the sites TAC, HFD, RGL, and BSD showing the largest improvements, thereby highlighting the importance of the UK network expansion.

And (in the supplements)

We show the model–measurement agreement in Tab. S7 (r, RMSE, and bias) and with a Taylor diagram (Fig. S3). The Taylor diagram summarizes the match between observed and simulated values by

showing: (i) the Pearson correlation coefficient (r), represented by the azimuthal angle, and (ii) the normalized standard deviation, represented by the radial distance from the origin. Each colored arrow represents a station, where the tail represents the a priori and the head the a posteriori simulated mole fractions. The length of each arrow represents the difference between the a priori and a posteriori modeled mole fractions with respect to the observed mixing ratios, indicating the correction made by the inversion. Overall, the Taylor diagram highlights the most pronounced statistical improvements at the UK stations TAC, HFD, RGL, and BSD. The Pearson correlation coefficient increased by about 0.1 at TAC and HFD, and by around 0.05 at RGL and BSD, averaged over their respective operating years. These stations also show substantial reductions in both bias and RMSE (see Tab. S7). The Taylor plot further shows notable improvements at ZEP, MHD, and the mountain station ZSF in individual years. The other European mountain stations, JFJ and CMN, also exhibit modest increases in r values (around 0.02) but pronounced reductions in bias. Finally, Fig. S3 clearly differentiates between stations inside and outside Europe, with non-European stations showing little to no improvement in r values. However, some display reductions in bias and RMSE, most likely due to the improved baseline (Tab. S7).

Specific comments:

Abstract: Usually, an abstract should start with a problem statement and then outline what the current work adds to understanding/solving the problem. Hence, I suggest adding one sentence discussing the importance of SF6 (potentially very similar to the one in introduction).

We changed to:

Sulfur hexafluoride ($SF_6$) is a highly potent and long-lived greenhouse gas whose atmospheric concentrations are increasing due to human emissions. In this study, we determine European $SF_6$ emissions from 2005 to 2021 using a large ensemble of atmospheric inversions.

Introduction 1[st] and 3[rd] paragraph: To me it would make sense to move the 3[rd] paragraph up and continue the 1[st] paragraph with it (atmospheric importance of SF6) and then come to the usage/sources.

Yes, thank you. We moved up the 3[rd] paragraph.

L89ff: It would be good to understand how different these observational datasets are in the end. Maybe add a table giving number of observations for each of the eight groups. Also, the distinction between 1 and 2 is not so clear, since Fig 1 only shows European sites. Which global sites were used that would provide any reasonable constraint on European emissions. The description implies that all your inversion domains are always global. Is this correct? Should be highlighted somewhere besides mentioning the previous method paper.

Yes, we agree that our initial description requires more details. We added more information on the datasets (see answer to third general comment).

No, the inversions themselves are not global, the emissions are optimized only in the European domain (see emissions grid), or in other terms: the state vector includes only emissions inside the European domain (except for baseline optimization factors that are also included). However, the LPDM simulations were performed globally (with a high-resolution nest at the European domain). This was done to (i) model the relationship between all observations (also those outside Europe) and emissions inside the inversion domain, (ii) model the trajectory endpoints for the baseline calculation, and (iii) calculate emissions contributions from outside the inversion domain that occurred during the LPDM

simulation period. The latter must be added to the baseline. So, although the LPDMs are run globally, the inversions themselves are regional. We add a sentence to make this clearer:

Although we also use observations from outside the European domain and perform FLEXPART simulations globally (with a European nest), the inversions are regional; that is, emissions are optimized only within Europe.

L131ff: The GAINS inventory is mentioned without a reference, only link to previous study by Vojta et al. 2024. Please clarify if the inventory is available at two different resolutions, as it seems, and why you chose two different re-gridding strategies. I suppose the second is simply a smoothing of the inventory, but the rational behind that should be explained.

We changed the description to make it clearer:

We created two a priori emission fields based on the GAINS inventory (Purohit and Höglund-Isaksson, 2017), which is detailed in Vojta et al. (2024). The inventory is available at 0.5° resolution globally and at 0.1° resolution for a European subset covering the EU-27, Iceland, Norway, Switzerland, and the UK. For the first field (GS), we re-gridded the global 0.5° inventory to 0.25° over the European domain by interpolation. For the second field (GS-HR), we used the higher-resolution European dataset, aggregated it to 0.25°, and combined it with the global dataset. While both fields thus share the same resolution (0.25° over Europe), the information content differs: GS is interpolated from coarser data, whereas GS-HR retains detail from the original high-resolution European inventory.

Sec. 2.3: What is the reason for using the population and the night-light proxies that seem to have an extremely similar distribution (judging from Fig.3). Instead, a proxy with very different distribution could have been interesting (e.g., uniform within each country or reflecting the electric grid). In addition: what is the reference year for the population and night-light data?

Yes, the population- and night-light–based priors are more similar to each other than the priors from different inventories; however, there are still notable differences between them (see, for example, Fig. B2). We agree that using a fundamentally different spatial distribution, such as a uniform prior, could also have been interesting to investigate. However, one has to keep in mind that the a priori errors in our setup are also defined based on the gridded emission values, and adopting a uniform prior would require substantial modifications to the inversion configuration, making direct comparisons difficult.

Sec. 2.3, UNFCCC-based prior: How are non-reporting countries treated? Several Eastern European Countries seem to have extremely low emissions compared to GAiNS and EDGAR. On the other hand, north African countries seem to have large emissions

The treatment of non-reporting countries is described in a footnote: "[2]Emissions of non-Annex I countries, that fall within our inversion domain but are not further investigated, were estimated proportionally to their national electricity generation as described in Vojta et al. (2024)."

L179: Does this take the posterior covariance between different grid cells into account?

We hope we understand the reviewer's question here correctly: The aggregation of the a posteriori emissions for individual countries does not take posterior covariances between different grid cells into account (in contrast to the aggregation of a posteriori errors, where the correlations should be included)

Sec. 2.5: Although, there is information in Tab. 1, I am missing a short description of the reference inversion in the text. A short explanation how parameter values were chosen for the reference inversion would also be helpful.

We added:

For the reference inversion, parameter choices were informed by a set of preliminary runs, in which we evaluated chi-squared statistics, and by values reported from previous studies.

L204: Please comment on the question of how much omitting off-diagonal covariance may impact the inversion results. In addition, to simply scaling the diagonal elements of the covariance matrix, changing its structure is usually another important sensitivity test.

We added a footnote: Omitting the off-diagonal elements of the observation error covariance matrix could potentially lead to an underestimation of the total observation uncertainty, resulting in an over-weighting of observations. This could especially be relevant for high-frequency observations, driving results further away from the a priori emissions. However, we reduce this risk by averaging the observations and by verifying through Chi-squared statistics that the assumed uncertainties remain consistent with the data.

Observation and prior uncertainty: Others have tested more objective ways of setting observation and prior uncertainties; for example, by evaluating Chi-square statistics of the cost function. Here, values for the observational uncertainty within a range of a factor of 5 were tested, the smallest of which almost certainly are too small given the analytical uncertainty alone. Too small values of the observational uncertainty are known to lead to over-fitting of results and can mostly be avoided from the beginning. Including ensemble inversions with unrealistic settings will artificially increase posterior uncertainties as explored here through the ensemble spread. How did the two 'variable' estimates of observational uncertainty compare to the fixed values.

Yes, we agree with that comment. The choice of the reference inversion and the tested uncertainty ranges was actually informed by evaluating Chi-square statistics of the cost function, which we now explain in the revised manuscript. We additionally compute and discuss Chi-squared statistics for the uncertainty sensitivity tests to refine and justify the chosen parameter ranges. Furthermore, we agree that implausible uncertainty values should be excluded from the outset. Accordingly, we revised the parameter ranges and recalculated the inversions, sampling the parameter space from a normal distribution rather than a uniform distribution (see response to the second main comment).

"The two approaches that account for spatial variability in the model uncertainty generally fall within the range of constant-error settings, although they show a slightly different pattern for some periods."

"The chi-values related to the spatial varying uncertainties were 0.80 (standard deviation, a posteriori distribution), and 0.31 (RMSE, a priori distribution). Indeed, using the RMSE of the a priori distribution is expected to be an overestimation, as it also reflects systematic mismatches and biases in the a priori emissions rather than purely observational uncertainty."

L218: 'global field at once' Do you mean global field as a whole? Or in other words: one scaling factor for the whole global field? Please rephrase.

Yes, one global scaling factor, we rephrased to:

We also run inversions where we optimize one scaling factor for the whole global field.

L254f: An alternative way to look at the information content in a Bayesian inversion, would be the exploration of the averaging kernel, which should be closer to one in areas where the posteriori result is mostly informed by the observations and closer to zero where information comes mainly from the prior. Most likely, the averaging kernel will be similar to the uncertainty reduction, but I wonder if areas like Israel and Russia would show up in it as well.

We agree that the exploration of the averaging kernel would be another interesting way to look at the information content. However, given that this study is already quite extensive, we prefer to stay with the error reduction.

L287: One advantage of the presented ensemble approach could be to give non-Gaussian uncertainties of the posterior. However, it seems that uncertainty is given here as the standard deviation of the ensemble instead of 2.5 % and 97.5% percentile range. How do these two estimates compare?

The two estimates are generally very similar; however, we agree that non-Gaussian uncertainties could be an advantage of the ensemble approach. Therefore, we report uncertainties with the 2.5 % and 97.5% percentile range in the revised manuscript, as suggested by the reviewer.

Fig. 6 and others: It is somewhat difficult to distinguish the vertical lines for network extension from the grid lines. Could the grey lines for network extension be given as dashed lines instead?

Yes, of course, thank you.

L312: Consider starting new paragraph after 'inventory.' New point and figure.

Yes, thank you -> done

L316: Comment on the fact that the reduction on Northern Germany is quite similar to the reduction in the UK (about a factor of 3).

Yes, great point. We added:

Note that this decrease in emissions is comparable to the reduction observed in the UK.

L351f: I find this conclusion too far fetched. Seeing the range of possible inversion results from one system by changing some of the critical parameters and not knowing any of these, or similar, parameters used by Manning et al. (2022), the good agreement may just be by chance. What would be needed for the suggested conclusion would be a direct switch of transport models between the two inversion systems and an alignment of input parameters to cross-check the results. Hence, I suggest to somewhat weaken the conclusion at this point.

Yes, we generally agree with this statement. However, especially in the case of the UK, seeing the stability of the results across all the different settings, we find it hard to believe that the similarity to the results from Manning et al. (2022) could be just by chance.  But of course, the reviewer is right that a direct switch of transport models between the two inversion systems and an alignment of input parameters to cross-check the results would be needed to prove that. We weakened the statement to:

Nevertheless, the excellent agreement of the emissions in the UK and northwestern Europe (Fig. 6a and Fig. A3) with those reported by Manning et al. (2022), particularly after the network expansion, might suggest that, with a dense monitoring network, inversion results remain stable even when these factors change.

L515f: Is it worth discussing these uncertain results with a dedicated Figure?

With this figure, we want to show that despite the large increments and the notable error reduction, results remain uncertain in this region, as the reviewer points out. We therefore think it is worthwhile showing this figure.

L525ff: This is connected to my general comment on observational data availability and model performance. If I am not mistaken, Monte Cimone only provided reliable SF6 data until the end of 2017 (WDCGG). Then in 2023 a new Medusa-GCMS was installed. For the period between there is very little constraint on Italian emissions. Hence, the results, as discussed in this section, need to acknowledge the observational data availability.

Yes, the reviewer is perfectly right: Monte Cimone provided data until 2017, which gives a very good explanation for the results. Thank you! We added:

This divergence is likely related to the fact that Monte Cimone provided observations only until 2017, after which constraints on Italian emissions were substantially reduced.

L578: I suppose this statement refers to larger a priori uncertainty for individual grid cells? Or does the domain total uncertainty change with correlation length?

Yes, the aggregated uncertainty increases with correlation length, but the statement might generally be confusing, and we changed it to:

However, this should not be interpreted as an indication of a superior inversion quality. It reflects the broader spatial distribution of observational information rather than an actual improvement in the ability of the inversion to constrain emissions.

L598: I would rather be skeptical about this statement. I would think that very distant observations may introduce large biases in posterior estimates for two reasons: 1) long transport is usually connected to large uncertainties, 2) infrequent observation of the same source area when source and receptor are very distant.

We changed to:

This indicates that distant observations have little influence on emissions in countries that are well observed, but can affect emissions in areas that are less well covered.

L603: Should MCN be CMN?

Yes, thank you!

L620f: If posterior emissions change little with applied observation uncertainties, how does posterior uncertainty change? Smaller posterior uncertainty should be seen for smaller observation uncertainty unless observation uncertainty is generally underestimated.

Yes, posterior uncertainty decreases with smaller observation uncertainty – see for instance the error reduction when comparing an observation uncertainty of 0.02 ppt and 0.1 ppt

[Figure]

Fig A2 and I1: The colours for individual inversions only distract. Unless there is another grouping that should be indicated by the colours, I suggest dropping the colours and represent individual runs by gray lines

Done!